# Uncertainties in the observation and simulation of global speciated atmospheric mercury deposition to terrestrial surfaces

**Lei Zhang[1,2,*], Peisheng Zhou[1], Shuzhen Cao[1], and Yu Zhao[1,2]**

[1] School of the Environment, Nanjing University, 163 Xianlin Avenue, Nanjing, Jiangsu 210023, China

[2] State Key Laboratory of Pollution Control and Resource Reuse, Nanjing University, 163 Xianlin Avenue, Nanjing, Jiangsu 210023, China

*Correspondence to:* Lei Zhang (lzhang12@nju.edu.cn)

**Abstract.** One of the most important processes in the global mercury (Hg) biogeochemical cycling is the deposition of atmospheric Hg, including gaseous elemental mercury (GEM), gaseous oxidized mercury (GOM), and particulate-bound mercury (PBM), to terrestrial surfaces. Results of wet, dry, and forest Hg deposition from global observation networks, individual monitoring studies, and observation-based simulations have been reviewed in this study. Uncertainties in the observation and simulation of global speciated atmospheric Hg deposition to terrestrial surfaces have been systemically estimated based on assessment of commonly used observation methods, campaign results for comparison of different methods, model evaluation with observation data, and sensitivity analysis for model parameterization. The uncertainties of GOM and PBM dry deposition measurements come from the interference of unwanted Hg forms or incomplete capture of targeted Hg forms, while that of GEM dry deposition observation originates from the lack of standardized experimental system and operating procedure. The large biases in the measurements of GOM and PBM concentration and the high sensitivities of key parameters in resistance models lead to high uncertainties in GOM and PBM dry deposition simulation. Non-precipitation Hg wet deposition could play a crucial role in alpine and coastal regions, and its high uncertainties in both observation and simulation affect the overall uncertainties of Hg wet deposition. The overall uncertainties in the observation and simulation of the total global Hg deposition were estimated to be ±(30–50) % and ±(50–70) %, respectively, with the largest contributions from dry

deposition. According to the results from uncertainty analysis, future research needs were recommended, among which global Hg dry deposition network, unified methods for GOM and PBM dry deposition measurements, quantitative methods for GOM speciation, campaigns for comprehensive forest Hg behavior, and more efforts on long-term Hg deposition monitoring in Asia are the top priorities.

## 1 Introduction

Mercury (Hg) is a global pollutant, characterized by its neurotoxicity, persistency and bioaccumulation effect. It undergoes regional or global long-range transport via atmospheric circulation, deposition to local or remote areas, methylation in ecosystems, and accumulation through food chain, posing high risks to human health and the environment (Obrist et al., 2018). Hg in the atmosphere has three major forms: gaseous elemental mercury (GEM), gaseous oxidized mercury (GOM), and particulate-bound mercury (PBM). The sum of the three Hg forms is named total mercury (TM). GOM and PBM are also known as reactive mercury (RM). GEM is the predominant form of atmospheric Hg (>90 %) with a long residence time of several months to over one year due to its chemical inertness and low solubility. GOM used to be estimated to account for less than 1 % of atmospheric Hg, which is easily scavenged by wet deposition, resulting in a short residence time of hours to days (Schroeder and Munthe, 1998; Lindberg et al., 2007). However, recent studies (Lyman et al., 2010; Gustin et al., 2013; McClure et al., 2014; Gustin et al., 2015) show that there could be a significant underestimation of GOM due to the low capture efficiency of the KCl denuder method adopted by most observation sites in the presence of ozone or moisture. PBM (<10 % of atmospheric Hg) stays in the air for days to several weeks depending on particle size before scavenged by dry or wet deposition (Schroeder and Munthe, 1998; Lindberg et al., 2007; Ci et al., 2012; Fu et al., 2012; Zhang et al., 2016a).

Deposition is one of the most important processes in global Hg cycling, leading to the sink of atmospheric Hg (Obrist et al., 2018). According to the Global Mercury Assessment 2018 (UN Environment, 2019), the annual Hg deposition to land and freshwater is estimated to be 3600 t. Atmospheric Hg deposition can be broadly divided into wet and dry deposition. Hg wet deposition is mostly in the form of precipitation (rain, snow, etc.), with non-negligible contribution from non-

precipitation forms (cloud, fog, dew, frost, etc.). Hg dry deposition is highly related to the underlying surfaces, including forest canopies, grasslands, wetlands, agricultural fields, deserts, background non-vegetated soils, contaminated sites, etc. (Zhang et al., 2009). Forest canopy is regarded as an important sink of atmospheric Hg for its special forms of deposition, litterfall and throughfall (Gustin et al., 2008). Litterfall is a form of indirect Hg dry deposition through foliar uptake of atmospheric Hg, and throughfall includes wet-deposited Hg above the canopy and a portion of dry-deposited Hg washed off from the canopy (Wright et al., 2016). Hg deposition through litterfall has recently been drawn much attention to by the study of Wang et al. (2016a). The sum of litterfall and throughfall is regarded as the total Hg deposition in forest canopies.

Significant efforts have been made in the past decade for quantifying atmospheric Hg deposition through both direct observations and model simulations, especially on dry deposition (Lyman et al., 2009; Zhang et al., 2009; Holmes et al., 2011; Lai et al., 2011; Castro et al., 2012; Gustin et al., 2012; Peterson et al., 2012; L. Zhang et al., 2012; Fang et al., 2013; Sather et al., 2013; Lynam et al., 2014; Sather et al., 2014; Huang and Gustin, 2015a; Weiss-Penzias et al., 2016a; Zhang et al., 2016b; Hall et al., 2017; Sprovieri et al., 2017). Yet large uncertainties still exist due to limitations of current methods for Hg deposition measurements and modeling (Gustin et al., 2015). The purpose of this paper is to give an overview of the uncertainties in the observation and simulation of global speciated atmospheric Hg deposition to terrestrial surfaces. In this paper, we investigated results from the observation and simulation of global Hg deposition, reviewed methods adopted for Hg deposition measurements and modeling, estimated the uncertainties of different methods for different Hg deposition forms, and summarized the overall uncertainty level of global Hg deposition.

## 2 Observation-based estimation of global Hg deposition

### 2.1 Wet deposition

Precipitation is the major form of Hg wet deposition. There have been several observation networks of Hg wet deposition through precipitation. The Global Mercury Observation System (GMOS) is so far the only global scale network covering the northern hemisphere, the tropics, and the southern hemisphere (Sprovieri et al., 2017). The Mercury Deposition Network (MDN) of the National Atmospheric Deposition

Program (NADP) in North America is the earliest continental scale network
specifically for Hg deposition (Prestbo and Gay, 2009; Weiss-Penzias et al., 2016a).
Hg wet deposition is also monitored in the European Monitoring and Evaluation
Programme (EMEP) for Europe (Tørseth et al., 2012; Bieser et al., 2014). A new
Asia–Pacific Mercury Monitoring Network has recently been established (Obrist et
al., 2018). Figure 1 summarizes the global distribution of the observed Hg wet
deposition fluxes based on results from both these global or regional networks and
individual studies.
Sprovieri et al. (2017) reported a 5-year record (2011–2015) of Hg wet deposition
at 17 selected GMOS monitoring sites, which provided a global baseline of the Hg
wet deposition flux including regions in the southern hemisphere and tropical areas.
The average Hg wet deposition fluxes in the northern hemisphere, the tropics, and the
southern hemisphere were 2.9 (0.2–6.7), 4.7 (2.4–7.0), and 1.9 (0.3–3.3) $\mu g\ m^{-2}\ yr^{-1}$,
respectively. The MDN network has a much longer history dating back to the 1990s.
Weiss-Penzias et al. (2016a) analyzed records from 19 sites in the United States (U.S.)
and Canada between 1997 and 2013, and discovered trends of Hg concentration in
wet deposition, with the early time period (1998–2007) producing a significantly
negative trend (−1.5±0.2 % $yr^{-1}$) and the late time period (2008–2013) a flat slope (not
significant). Therefore, the MDN data of 136 sites for the time period of 2008–2015
(http://nadp.slh.wisc.edu/mdn) were used in Figure 1 to represent the recent
background Hg wet deposition level in North America. Fu et al. (2016a) summarized
wet deposition measurements from 7 monitoring sites in China. Hg wet deposition
fluxes at rural sites in forest and grassland were averagely 6.2 and 2.0 $\mu g\ m^{-2}\ yr^{-1}$,
respectively, while the flux at an urban site was as high as 12.6±6.5 $\mu g\ m^{-2}\ yr^{-1}$.
Overall, East Asia has the highest wet deposition flux (averagely 16.1 $\mu g\ m^{-2}\ yr^{-1}$),
especially in the southern part of China where the GEM concentration level is
relatively high (Fu et al., 2008; Guo et al., 2008; Wang et al., 2009; Fu et al., 2010a;
2010b; Ahn et al., 2011; Huang et al., 2012b; Seo et al., 2012; Huang et al., 2013;
Sheu and Lin, 2013; Marumoto and Matsuyama, 2014; Xu et al., 2014; Zhu et al.,
2014; Huang et al., 2015; Zhao et al., 2015; Han et al., 2016; Fu et al., 2016a; Ma et
al., 2016; Nguyen et al., 2016; Qin et al., 2016; Sommar et at., 2016; Cheng et al.,
2017; Chen et al., 2018; Lu and Liu, 2018). North America has an average Hg wet
deposition flux of 9.1 $\mu g\ m^{-2}\ yr^{-1}$, and exhibits a descending spatial profile from the

southeastern part to the northwestern part, which is consistent with the distribution of

the atmospheric Hg concentration (L. Zhang et al., 2012; Gichuki and Mason, 2014;

Lynam et al., 2017). Europe has the lowest Hg wet deposition level (averagely 3.4 μg

m$^{-2}$ yr$^{-1}$) according to the available observation and simulation data (Connan et al.,

2013; Bieser et al., 2014; Siudek et al., 2016). Observation data for the tropics and the

southern hemisphere are scarce with large uncertainties (Wetang'ula, 2011; Gichuki

and Manson, 2013; Sprovieri et al., 2017). The one exceptional tropical site with a

wet deposition flux of 16.8 μg m$^{-2}$ yr$^{-1}$ is in Kenya while the other sites in the tropics

are all in Mexico (Wetang'ula, 2011; Hansen and Gay, 2013). The two sites in the

southern hemisphere with annual precipitation of over 4000 mm are in Australia and

have wet deposition fluxes of 29.1 and 18.2 μg m$^{-2}$ yr$^{-1}$, respectively (Dutt et al.,

2009). Seen from the bottom part of Figure 1, Hg wet deposition flux is not

significantly correlated with elevation.

Hg wet deposition on different terrestrial surface types were investigated in this

study. As shown in Figure 2, the average Hg wet deposition flux follows the

ascending sequence of barren areas, grasslands, croplands, savannas, and urban areas.

The wet deposition level has a strong correlation with precipitation on these surfaces.

The "water" surfaces here refer to the terrestrial surfaces near water, e.g., coastal,

offshore, and lakeside sites. The near-water surfaces and forest canopies have lower

Hg wet deposition levels than the other surfaces at a similar amount of precipitation.

In other words, the Hg concentrations in precipitation for these two types of surface

types are lower (by 20–30 %) than for the other types. This is possibly related to non-

precipitation Hg wet deposition (e.g., cloud, fog, dew, and frost). Fog or cloud Hg

deposition is not yet considered in the global Hg wet deposition observation network.

However, studies (Stankwitz et al., 2012; Weiss-Penzias et al., 2016b; Gerson et al.,

2017) have shown that cloud and fog water have higher Hg concentration than rain

water in the same region, and cloud and fog could have a remarkable contribution to

Hg wet deposition in high-elevation forests and near-water surfaces. Cloud and fog

scavenging of reactive Hg (GOM and PBM) could result in lower Hg concentration in

precipitation.

Studies on non-precipitation Hg wet deposition (e.g., cloud, fog, dew, and frost) are

very limited so far. Stankwitz et al. (2012) and Gerson et al. (2017) found the average

cloud Hg deposition fluxes of two North American montane forests to be 7.4 and 4.3

µg m$^{-2}$ yr$^{-1}$, respectively, equivalent to rainfall Hg deposition. In California coastline,
fog Hg deposition, with only 2 % volume proportion, accounts for 13 % of the total
wet deposition (Weiss-Penzias et al., 2016b). Converse et al. (2014) found the annual
dew and frost Hg deposition at a high-elevation meadow in the U.S. to be about 0.12
µg m$^{-2}$ yr$^{-1}$, 2–3 orders of magnitude smaller than wet deposition. More standardized
method are in urgent need for non-precipitation Hg wet deposition measurements.
**2.2   Dry deposition**
Observation-based estimation of Hg dry deposition consists of two types, direct
measurements of speciated Hg dry deposition fluxes and model simulations based on
observation of speciated atmospheric Hg concentrations. Figure 3 shows the global
distribution of the GOM, PBM and GEM dry deposition fluxes from observation-
based estimation (either direct observation of dry deposition or simulation based on
Hg concentration observation). The global Hg dry deposition network is very
immature compared to the wet deposition network due to the inconsistency in
methods for estimation. GOM dry deposition fluxes were either measured by the
surrogate surface methods or simulated based on GOM concentration measurements.
PBM dry deposition fluxes were mainly estimated from the measurements of total or
size-resolved PBM concentrations. GEM dry deposition fluxes were measured by
different types of methods, the surrogate surface methods, the enclosure methods, and
the micrometeorological methods.

Most studies on GOM dry deposition were conducted in North America and

Europe, among which direct observations of GOM dry deposition are mainly from
North America (Lyman et al., 2007; Lyman et al., 2009; Weiss-Penzias et al., 2011;
Lombard et al., 2011; Castro et al., 2012; Gustin et al., 2012; Peterson et al., 2012;
Zhang et al., 2012; Sather et al., 2013; Bieser et al., 2014; Sather et al., 2014; Wright
et al., 2014; Huang and Guatin, 2015a; Enrico et al., 2016; Han et al., 2016; Zhang et
al., 2016b; Huang et al., 2017). Regardless of the estimating methods, the average
GOM dry deposition flux in North America (6.4 µg m$^{-2}$ yr$^{-1}$) is higher than in Europe
(3.0 µg m$^{-2}$ yr$^{-1}$). There have been very few studies on GOM dry deposition in Asia.
Han et al. (2016) used knife-edge surrogate surface (KSS) samplers with quartz filters
to measure GOM dry deposition at a remote site in South Korea, and found an
average GOM dry deposition flux of 4.78 µg m$^{-2}$ yr$^{-1}$. A significant correlation
($R^2$=0.532, p<0.01) was found between the elevation and the GOM dry deposition

flux (Figure 4). Huang and Gustin (2015a) found that measured dry deposition of GOM was significantly high at sites over 2000 m above sea level, and attributed it to high GOM concentrations at high elevation and atmospheric turbulence. Significant discrepancies were found between the GOM dry deposition fluxes from direct observations and from model simulations based on measurements of GOM concentrations (Figure 5).

Due to the severe particulate matter (PM) pollution in East Asia, many independent size-resolved PM measurements were conducted in recent years with analysis of Hg in PM accordingly. Results from size-resolved PBM analysis and PBM dry deposition models show that East Asia has a much higher average of PBM dry deposition flux (45.3 $\mu g\ m^{-2}\ yr^{-1}$) than North America (1.1 $\mu g\ m^{-2}\ yr^{-1}$) with coarse-particle PBM dry deposition not considered (Fang et al., 2012a; Fang et al., 2012b; Zhu et al., 2014; Zhang et al., 2015; Huang et al., 2016; Guo et al., 2017). Studies (Fang et al., 2012a; Zhu et al., 2014) have shown that Hg in coarse particles accounts for a large proportion of the total PBM, which was previously neglected, because PBM measured by the Tekran system only considers fine particles. Therefore, the PBM dry deposition could be generally underestimated.

Although large uncertainties still exist in the methods for GEM dry deposition measurements, it should be noted that GEM dry deposition is non-negligible compared to GOM and PBM. The average GEM dry deposition is lower in Europe (4.3±8.1 $\mu g\ m^{-2}\ yr^{-1}$) while higher in North America (5.2±15.5 $\mu g\ m^{-2}\ yr^{-1}$) with more variation (Castelle et al., 2009; Baya and Heyst, 2010; Converse et al., 2010; Miller et al., 2011). The four Asian sites using micrometeorological methods all show negative values (−36.3±19.6 $\mu g\ m^{-2}\ yr^{-1}$), indicating the role of East Asia as a net emission source rather than a net deposition sink (Luo et al., 2014; Luo et al., 2016; Ci et al., 2016; Yu et al., 2018). However, the GEM dry deposition observation in Asia is still very limited. Agnan et al. (2016) and Zhu et al. (2016) made detailed summaries of campaign-based GEM dry deposition observations, and addressed the importance of natural Hg emission sources.

Figure 6 exhibits the dry deposition fluxes of GOM, PBM and GEM for different terrestrial surface types. As shown in Figure 6a, high GOM dry deposition levels were found for grasslands (mainly alpine meadows) and savannas. This is probably because of the enhanced Hg oxidation process at high elevations with more halogen free

radicals or more intensive solar radiations. Urban areas also have high GOM dry
deposition fluxes due to high GOM concentrations. The low GOM dry deposition
fluxes on moist surfaces (near-water surfaces and croplands) might be partially
because of fog and dew scavenging (Malcolm and Keeler. 2002; Zhang et al., 2009).
The PBM dry deposition flux is high on surfaces with high human activities (urban
areas and croplands) and low in vegetative areas, implying the heavier PM pollution
in urban and rural areas than in remote areas (Figure 6b). Short-term observation of
GEM dry deposition shows high fluctuation. Therefore, we summarized model
estimations and one annual observation dataset (L. Zhang et al., 2012; Bieser et al.,
2014; Zhang et al., 2016b; Enrico et al., 2016), and found that the GEM dry
deposition does not only depend on GEM concentration, but also on the air–soil Hg
exchange compensation point (Luo et al., 2016). Regarding the annual air–surface Hg
exchange, instead of an important natural source, forests tend to be a net sink of
atmospheric Hg (Figure 6c).
**2.3  Forest deposition**
Hg deposition in forests is mainly in the forms of litterfall and throughfall. Wang et
al. (2016a) made a comprehensive assessment of the global Hg deposition through
litterfall, and found litterfall Hg deposition an important input to terrestrial forest
ecosystems ($1180\pm710$ Mg $yr^{-1}$). South America was estimated to bear the highest
litterfall Hg deposition ($65.8\pm57.5$ μg $m^{-2}$ $yr^{-1}$) around the world. This was partially
because some studies were conducted in the Amazonian rainforest (Fostier et al.,
2015), mainly semi-deciduous or evergreen tropical forest, which account for over
40% litterfall deposition globally (Shen et al., 2019). Another reason was that some
sampling sites were very close to large cities or polluted areas, which could lead to
more Hg accumulation (Teixeira et al., 2012; Buch et al., 2015; Teixeira et al., 2017;
Fragoso et al., 2018). There have been numerous forest Hg deposition studies in the
recent decade in East Asia with the second highest average litterfall Hg deposition
flux ($35.5\pm27.7$ μg $m^{-2}$ $yr^{-1}$). The forest type varies among different studies, but East
Asia has much higher Hg concentrations in litterfall ($42.9$–$62.8$ ng $g^{-1}$) compared to
other regions (Wan et al., 2009; Wang et al., 2009; Fu et al., 2010a; Fu et al., 2010b;
Gong et al., 2014; Luo et al., 2016; Ma et al., 2015; Han et al., 2016; Fu et al., 2016a;
Ma et al., 2016; Wang et al., 2016b; Zhou et al., 2016; Zhou et al., 2017). Lower
levels of litterfall Hg deposition fluxes were found in North America ($12.3\pm4.9$ μg $m^{-2}$
$yr^{-1}$) and Europe ($14.4\pm5.8$ µg m$^{-2}$ yr$^{-1}$) (Larssen et al., 2008; Obrist et al., 2009;
Fisher and Wolfe, 2012; Juillerat et al., 2012; Obrist et al., 2012; Risch et al., 2012;
Benoit et al., 2013; Navrátil et al., 2014; Gerson et al., 2017; Risch et al., 2017; Risch
and Kenski, 2018). According to Risch et al. (2017), the litterfall Hg deposition flux
in the eastern U.S. decreased year by year during 2007–2014 with a declining rate of
0.8 µg m$^{-2}$ yr$^{-1}$. From 2007 to 2009 the decrease occurred more rapidly due to the Hg
emission control strategies during this period of time. The litterfall Hg deposition flux
and the Hg concentration in litterfall are shown in Figure 7. In general, evergreen
forests have higher litterfall Hg concentrations than deciduous forests due to longer
accumulation time (Wright et al., 2016). Evergreen broadleaf forests have not only
high litterfall Hg concentrations but also high litterfall rates (Shen et al., 2019), and
consequently bear high litterfall Hg deposition. Comparing the levels of wet, dry, and
litterfall Hg depositions in forests, litterfall markedly takes the lead, especially for
evergreen broadleaf forests. This is consistent with the budget of global litterfall Hg
deposition developed by Wang et al. (2016a).

Most studies on Hg deposition in forests in North America use rainfall instead of

throughfall since dry deposition in North American forests has limited contribution
(Risch et al., 2017), while Asian studies found large discrepancy between throughfall
and rainfall Hg deposition fluxes ($32.9\pm18.9$ and $13.3\pm8.6$ µg m$^{-2}$ yr$^{-1}$, respectively),
indicating a high dry deposition level in Asian forests (Wan et al., 2009; Wang et al.,
2009; Fu et al., 2010a; Fu et al., 2010b; Luo et al., 2016; Ma et al., 2015; Han et al.,
2016; Fu et al., 2016a; Ma et al., 2016; Wang et al., 2016b; Zhou et al., 2016).
Litterfall and throughfall Hg deposition fluxes are equivalent. Wright et al. (2016)
summarized previous studies and reported the mean litterfall and throughfall Hg
deposition, respectively, 42.8 and 43.5 µg m$^{-2}$ yr$^{-1}$ in Asia, 14.2 and 19.0 µg m$^{-2}$ yr$^{-1}$
in Europe, and 12.9 and 9.3 µg m$^{-2}$ yr$^{-1}$ in North America.
**3    Uncertainties in Hg deposition observation**
**3.1    Uncertainties in the measurements of Hg wet deposition**
**3.1.1 Measurements of Hg wet deposition through precipitation**
Hg wet deposition through precipitation, mostly rainfall, is easier to measure than dry
deposition and usually more reliable. The rainfall Hg wet deposition flux is calculated
as follows (Zhao et al., 2018):
$$F_{\text{wet,rainfall}} = \sum_{i=1}^{n} C_i \cdot D_i \tag{1}$$

where $F_{\text{wet,rainfall}}$ is the total rainfall Hg wet deposition flux; $n$ is the number of
precipitation events during a certain period; $C_i$ is the total Hg concentration in
rainwater during Event $i$; and $D_i$ is the precipitation depth of Event $i$.
Both manual and automatic precipitation sample collectors were used in previous
studies (Fu et al., 2010a; Gratz and Keeler, 2011; Marumoto and Matsuyama, 2014;
Zhu et al., 2014; Brunke et al., 2016; Chen et al., 2018). The collected water samples
are preserved with HCl or BrCl in cool and dark environment for up to one month in
case of potential wall loss and photo-induced reduction of Hg (EPA Method 1631E;
Sprovieri et al., 2017). The total Hg concentration in water samples is then analyzed
by oxidation, purge and trap, and cold vapor atomic fluorescence spectrometry
(CVAFS) following EPA Method 1631, which allows the relative percent difference
(RPD) between field duplicates to be no more than 20 %. GMOS reported their
ongoing precision recovery (OPR) for every 12 samples to be generally within 93–
109 % (Sprovieri et al., 2017). The RPD for MDN precipitation Hg analysis is
generally within 10 % according to the inter-laboratory comparisons in the external
quality assurance project (2015–2016) conducted by the United States Geological
Survey (USGS) for MDN. For individual studies (Fu et al., 2010a; Huang et al., 2015;
Zhao et al., 2018), the relative standard deviation (RSD) is also generally less than
10 %. Overall, the relative uncertainty in rainwater Hg concentration analysis is
estimated to be ±10 %.
Automatic precipitation sample collectors cover the lid automatically when it is not
raining to prevent potential contamination, while manual collectors require manually
placing collectors before precipitation events and retrieving them after events. The
measurements of precipitation volume by sample collectors also have uncertainties
(Wetherbee et al., 2017). Based on the USGS report (2015–2016) for MDN, the RSD
of the daily measured precipitation depth by electronically recording gauges was
within 7 %, which was close to an early study (Wetherbee et al., 2005). Therefore, the
relative uncertainty in precipitation depth measurements is estimated to be ±7 %.
The uncertainty of the precipitation Hg wet deposition flux can be calculated based
on the uncertainties of the rainwater Hg concentration and the measurement of
precipitation depth. The relative uncertainty of precipitation Hg wet deposition is
estimated to be ±12 % using the following equation:

$$\delta_F(\text{wet}) = \frac{U_F(\text{wet})}{F_{\text{wet}}} = \sqrt{\left(\frac{U_C}{C}\right)^2 + \left(\frac{U_D}{D}\right)^2} = \sqrt{\delta_C^2 + \delta_D^2} \qquad (2)$$

where $\delta_F(\text{wet})$ and $U_F(\text{wet})$ are the relative and absolute uncertainties of Hg wet deposition flux, respectively; $\delta_C$ and $U_C$ are the relative and absolute uncertainties of the total Hg concentration in precipitation water, respectively; and $\delta_D$ and $U_D$ are the relative and absolute uncertainties of the precipitation depth, respectively.

**3.1.2 Measurements of Hg wet deposition through cloud, fog, dew and frost**

Non-precipitation Hg wet deposition, e.g., cloud, fog, dew and frost, could account for a notable proportion of the total wet deposition in montane, coastal, arid, and semi-arid areas (Lawson et al., 2003; Sheu and Lin, 2011; Stankwitz et al., 2012; Blackwell and Driscoll, 2015b). Quantifying Hg in cloud or fog helps better understand the impact of long-range transport and local sources on global Hg cycling (Malcolm et al., 2003). The non-precipitation Hg deposition flux is calculated as follows:

$$F_{\text{wet,non-precipitation}} = \sum_{j=1}^{m} C_j \cdot D_j \qquad (3)$$

where $F_{\text{wet,non-precipitation}}$ is the non-precipitation Hg deposition flux; $m$ is the number of non-precipitation wet deposition events during a certain period; $C_j$ is the total Hg concentration in non-precipitation wet deposition water during Event $j$; and $D_j$ is the non-precipitation wet deposition depth of Event $j$.

Both active and passive collectors have been used to collect cloud or fog water (Lawson et al., 2003; Malcolm et al., 2003; Kim et al., 2006; Sheu and Lin, 2011; Schwab et al., 2016; Weiss-Penzias et al., 2018). The major uncertainty lies in the deposition depth. The deposition depth of cloud, fog, dew or frost is usually modeled based on meteorology (Converse et al., 2014; Katata, 2014). The fog deposition depth can be measured by standard fog collectors (SFC). The uncertainty of fog deposition depth measurements is mainly from the collecting efficiency of SFC depending on the wind speed, wind direction, or mesh types (Weiss-Penzias et al., 2016b; Fernandez et al., 2018). Montecinos et al. (2018) evaluated the collection efficiency of SFC to be up to 37 %. Therefore, there is extremely large uncertainty in the measurements of the fog deposition depth. Based on the fog deposition studies (Weiss-Penzias et al., 2016b; Fernandez et al., 2018; Montecinos et al., 2018), the overall uncertainty of non-precipitation Hg deposition flux observation is estimated to be ±300 %. Note that the true uncertainty range is not symmetric about the mean because some of the

underlying variables are lognormally distributed (Streets et al., 2005). A better
interpretation of "±300 %" might be "within a factor of 4".

**3.2   Uncertainties in the measurements of Hg dry deposition**

Direct measurements of the Hg dry deposition flux is technically challenging, large
uncertainties still exist in quantify Hg dry deposition accurately (Wright et al., 2016).
Three major categories of methods for direct Hg dry deposition measurements are the
surrogate surface methods, the enclosure methods, and the micrometeorological
methods (Zhang et al., 2009; Huang et al., 2014).

**3.2.1 Measurements of RM (GOM and PBM) dry deposition**

RM dry deposition flux is proportional to the corresponding RM concentration (Zhang
et al., 2009):
$$F_{\mathrm{dry,RM}} = v_d \cdot C_z \tag{4}$$
where $F_{\mathrm{dry,RM}}$ is the RM dry deposition flux; $C_z$ is the RM concentration at
reference height $z$; and $v_d$ is the dry deposition velocity.
Most of the RM dry deposition measurements used the surrogate surface methods
(Huang et al., 2014; Wright et al., 2016). The micrometeorological methods and the
enclosure methods were also adopted in some studies (Poissant et al., 2004; Zhang et
al., 2005; Skov et al., 2006), but not widely used due to the high uncertainties in the
measurements of GOM and PBM concentrations using the Tekran system. For the
surrogate surface methods, the RM dry deposition flux is determined using the
following equation (Huang et al., 2014):
$$F_{\mathrm{dry,SS}} = \frac{M}{A \cdot t} \tag{5}$$
where $F_{\mathrm{dry,SS}}$ is the Hg dry deposition flux using the surrogate surface methods; $M$
is the total Hg amount collected on the material during the sampling period; $A$ is the
surface area of the collection material; and $t$ is the exposure time.
Different surrogate surfaces were used to measure different RM forms. Mounts
with cation-exchange membranes (CEMs) are widely used for GOM dry deposition
measurements (Lyman et al., 2007; Lyman et al., 2009; Castro et al., 2012; Huang et
al., 2012a; Peterson et al., 2012; Sather et al., 2013). The down-facing aerodynamic
mount with CEM is considered to be the most reliable deployment for GOM dry
deposition measurements so far (Lyman et al., 2009; Huang et al., 2014). Knife-edge
surrogate surface (KSS) samplers with quartz fiber filter (QFFs) and dry deposition

plates (DDPs) were deployed for PBM dry deposition measurements (Lai et al., 2011; Fang et al., 2012b; Fang et al., 2013). However, these samplers are not well verified to reflect the deposition velocity of PBM, and hence not widely accepted. KCl-coated QFFs were used to measure the total RM (GOM+PBM) dry deposition, but failed to capture GOM efficiently (Lyman et al., 2009; Lai et al., 2011).

According to Eq. (4), the uncertainty of RM dry deposition comes from the uncertainties of RM concentration and dry deposition velocity. The uncertainty of RM concentration mainly originates from the interference of unwanted RM forms or incomplete capture of targeted RM forms. CEMs exhibited a GOM capture rate of 51–107 % in an active sampling system (Huang and Gustin, 2015b). The CEM mounts designed to measure only GOM dry deposition capture part of fine PBM (Lyman et al., 2009; Huang et al., 2014), while the KSS samplers with QFFs designed to measure only PBM dry deposition may also collect part of GOM (Rutter and Schauer, 2007; Gustin et al., 2015). Based on the RM concentration measurements and the surrogate surface method evaluations, the GOM concentration related uncertainty is estimated to be ±50 % (Lyman et al., 2009; Lyman et al., 2010; Gustin et al., 2012; Fang et al., 2013; Zhang et al., 2013; Huang et al., 2014). The design of the sampler (e.g., the sampler orientation, the shape of the sampler, variation in turbulence, low surface resistances, passivation, etc.) leads to the dry deposition velocity related uncertainty which is about ±50 % for GOM (Lyman et al., 2009; Lai et al., 2011; Huang et al., 2012a). Calculating based on the method described by Eq. (2), the overall uncertainty of GOM dry deposition observation is ±70 %. There is not enough information to quantify the overall uncertainty of PBM dry deposition observation in a similar way. Based on the distribution of daily samples in the study of Fang et al. (2012b), the overall uncertainty of PBM dry deposition measurements is assumed to be roughly ±100 % or within a factor of 2.

**3.2.2 Measurements of GEM dry deposition**

GEM has a low dry deposition velocity due to its mild activity, high volatility and low water solubility, and deposited GEM could re-emit into the atmosphere (Bullock et al., 2008; Fu et al., 2016b). Various methods have been applied to studies on air–surface GEM exchange, among which the enclosure methods and the micrometeorological methods were most commonly used (Zhang et al., 2009; Agnan et al., 2016; Zhu et al., 2016; Yu et al., 2018).

Micrometeorological methods are considered more reliable because of higher
temporal resolution and less interference from the microenvironment (Zhu et al.,
2016). With the high expenses of these methods, they are not as widely used as the
enclosure methods (Sommar et al., 2013a; Pierce et al., 2015). Micrometeorological
methods can be divided into the direct flux measurement methods and the gradient
methods. The most known one of the former is the relaxed eddy accumulation (REA)
method, while the latter include the aerodynamic (AER) method and the modified
Bowen-ratio (MBR) method (Zhang et al., 2009; Yu et al., 2018).
The REA method is based on sampling upward and downward moving eddies at
constant flow rates, which relies on an ultrasonic anemometer to detect the vertical
wind velocity and control the fast response valves. The GEM dry deposition flux
based on the REA method is calculated as follows (Sommer et al., 2013b):
$$F_{\mathrm{dry,REA}} = \beta \sigma_w (C_{\mathrm{down}} - C_{\mathrm{up}}) \tag{6}$$
where $F_{\mathrm{dry,REA}}$ is the GEM dry deposition flux measured by the REA method; $\beta$ is
relaxation coefficient; $\sigma_w$ is the standard deviation of the vertical wind speed; and
$C_{\mathrm{down}}$ and $C_{\mathrm{up}}$ are the downward and upward GEM concentration, respectively.
The REA method conducts upward and downward sampling at the same height,
eliminating the footprint difference and potential GEM formation and loss (Zhu et al.,
2016). Dual inlets were recommended and applied in recent studies due to advantages
of synchronous concentration determination (Sommar et al., 2013b; Zhu et al., 2015b;
Kamp et al., 2018; Osterwalder et al., 2016).
The gradient methods (AER and MBR) sample air at different height to get the
vertical GEM concentration gradient. For the AER method, the GEM dry deposition
flux is calculated using the following equation (Fritsche et al., 2008; Baya and Van
Heyst, 2010; Yu et al., 2018):
$$F_{\mathrm{dry,AER}} = K \frac{\partial C}{\partial z} \tag{7}$$
where $F_{\mathrm{dry,AER}}$ is the GEM dry deposition flux measured by the AER method; $K$ is
the turbulent transfer coefficient (Yu et al., 2018); and $\partial C/\partial z$ is the gradient of the
vertical GEM concentration.
For the MBR method, the GEM dry deposition flux is calculated based on the
theory that the flux ratio of GEM over the reference scalar (e.g., $H_2O$) is proportional
to their concentration gradients (Obrist et al., 2006; Converse et al., 2010):
$$F_{\text{dry,MBR}} = F_r \frac{\partial C_{\text{Hg}}}{\partial C_r} \qquad\qquad (8)$$
where $F_{\text{dry,MBR}}$ is the GEM dry deposition flux measured by the MBR method; $F_r$ is
the flux of the reference scalar; and $\partial C_{\text{Hg}}/\partial C_r$ is the concentration gradient ratio of
GEM over the reference scalar.
Enclosure methods rely on the conservation of mass and have been used for most
GEM flux measurements due to their relatively low costs, portability, versatility and
intuitive nature (Eckley et al., 2011; Sommar et al., 2013a; Sommar et al., 2013b;
Agnan et al., 2016; Zhu et al., 2016; Ma et al., 2018). The dynamic flux chamber
(DFC) method is the most commonly used enclosure method. A vacuum pump is
applied to draw air through a low Hg blank chamber at a constant flow, and the GEM
concentrations at the inlet and outlet of the chamber are measured sequentially by a
mercury analyzer coupled with a switchable valve. The GEM dry deposition flux is
calculated according to the following equation (Zhu et al., 2015a):
$$F_{\text{dry,DFC}} = \frac{Q(C_{\text{inlet}} - C_{\text{outlet}})}{A} \qquad\qquad (9)$$
where $F_{\text{dry,DFC}}$ is the GEM dry deposition flux measured by the DFC method; $Q$ is
the flushing flow rate; $C_{\text{inlet}}$ and $C_{\text{outlet}}$ are the GEM concentrations at the chamber
inlet and outlet, respectively; and $A$ is the area of the chamber footprint.
Different flushing flow rates, chamber designs and materials, as well as the lack of
standard operating protocol and blank correcting procedures, make it hard for
comparison between different studies (Eckley et al., 2010; Agnan et al., 2016;
Osterwalder et al., 2018). Choi and Holsen (2009) reported that the polycarbonate
DFC blocks most of the UV-B light from reaching the soil where $Hg^{2+}$ can be reduced
to $Hg^0$, and hence the GEM emission flux might be underestimated by at most 20 %.
A novel DFC, abbreviated as NDFC, was designed and utilized in recent studies (Lin
et al., 2012; Zhu et al., 2015a; Zhu et al., 2015b; Osterwalder et al., 2018). The GEM
dry deposition flux under atmospheric condition can be calculated based on the flux
measured by NDFC with the internal shear property precisely controlled and the
surface shear property (Lin et al., 2012).
The uncertainty of air–surface GEM exchange flux using the micrometeorological
methods were estimated to be up to ±30 % (Meyers et al., 1996; Lindberg et al., 2001;
Fritsche et al., 2008; Sommer et al., 2013a; Zhu et al., 2015b). The more widely used
enclosure methods have much higher uncertainties. Zhu et al. (2016) summarized
existing air–surface GEM exchange studies and found that the mean flux using
micrometeorological methods is higher than using DFCs by a factor of 2. Therefore,
the overall uncertainty of GEM dry deposition observation is estimated to be ±100 %.

### 3.3 Uncertainties in the measurements of Hg deposition in forests

In forest ecosystems, Hg dry and wet depositions are not easy to be distinguished
markedly, and litterfall and throughfall are commonly used to evaluate the total Hg
deposition (Wang et al., 2016a; Wright et al., 2016).

### 3.3.1 Litterfall Hg deposition measurements

Hg dry deposition in forests includes uptake of Hg by leaf stomata and cuticle, tree
bark, and underlying soil. Some of the deposited Hg in the soil may emit back into the
atmosphere and be captured by leaves, while some of the deposited Hg in leaves may
be translocated to branches, stems and roots (Risch et al., 2012). Litterfall Hg
deposition includes the remaining dry-deposited Hg in leaves and bark as well as the
captured Hg emitted from the soil (Blackwell and Driscoll, 2015a; Wright et al.,
2016). Litterfall Hg deposition flux is calculated as follows (Fisher and Wolfe, 2012):
$$F_{\text{litterfall}} = \frac{E_A \cdot C_l \cdot M_l}{A \cdot t} \qquad\qquad (10)$$
where $F_{\text{litterfall}}$ is the litterfall Hg deposition flux; $E_A$ is the litterfall trap area
expansion factor (note: leaves outside the area above the trap could fall into the trap
due to horizontal air fluctuation); $C_l$ is the Hg mass concentration in litterfall; $M_l$ is the
total dry weight of litterfall; $A$ is the litterfall trap area; and $t$ is the sampling time.
The Hg content in litterfall can be determined by thermal decomposition,
amalgamation, and cold vapor atomic absorption spectrophotometry (CVAAS)
following EPA Method 7473 (Richardson and Friedland, 2015; Fu et al., 2016a; Zhou
et al., 2017; Risch et al., 2017). Alternatively, the litterfall samples can be digested
into solution, and the extracted Hg in the solution can be analyzed following EPA
Method 1631E (Fu et al., 2010a; Fisher and Wolfe, 2012). The uncertainty in litterfall
Hg content analysis is about ±7 % according to the Litterfall Mercury Monitoring
Network developed by NADP (Risch et al., 2017) and individual studies (Benoit et
al., 2013; Ma et al., 2015; Zhou et al., 2016; Gerson et al., 2017). Litterfall samples
are collected during the leaf-growing or -falling seasons with litterfall traps or
collectors (Fisher and Wolfe, 2012). Total litterfall consists of leaves and needles,
woody material such as twigs and bark, and reproductive bodies such as flowers,

seeds, fruits, and nuts (Meier et al., 2006; Risch et al., 2012). The total litter mass collected by different samplers could cause a RSD of 16 % (Risch et al., 2012) and Risch et al., 2017). Therefore, the overall uncertainty of litterfall Hg deposition observation on a regular basis is estimated to be ±20 %. Moreover, based on the assumption that the total Hg concentration in litterfall is linearly accumulated during the growing season, some studies estimated litterfall Hg concentration by multiplying a scale factor, which may cause extra uncertainty (Bushey et al., 2008; Poissant et al., 2008; Fu et al., 2010a; Gong et al., 2014). Taking this into consideration, the overall uncertainty of litterfall Hg deposition observation is estimated to be ±30 %.

**3.3.2 Throughfall Hg deposition measurements**

Throughfall Hg deposition includes wet-deposited Hg above the canopy and a portion of dry-deposited Hg washed off from the canopy (Blackwell and Driscoll, 2015a; Wright et al., 2016). Throughfall Hg deposition flux is calculated as follows (Fisher and Wolfe, 2012):

$$F_{\text{throughfall}} = \frac{E_A \cdot C_t \cdot V_t}{A \cdot t} \tag{11}$$

where $F_{\text{throughfall}}$ is the throughfall Hg deposition flux; $E_A$ is the throughfall funnel area expansion factor; $C_t$ is the Hg mass concentration in throughfall; $V_t$ is the total volume of throughfall; $A$ is the throughfall funnel area; and $t$ is the sampling time.

Throughfall under canopy is usually collected using a passive bulk throughfall collector with a funnel connected a bottle for water storage (Wang et al., 2009; Fisher and Wolfe, 2012; Åkerblom et al., 2015) or collected as open-field rain collection if the environmental condition permits (Choi et al., 2008; Fu et al., 2010a; Fu et al., 2010b; Han et al., 2016). Attention should be paid to potential litterfall contamination and cloud or fog deposition influence at high elevation sites if the collector is not sheathed (Fisher and Wolfe, 2012; Wright et al., 2016). Throughfall samples are usually analyzed following EPA Method 1631E (Fisher and Wolfe, 2012). Therefore, throughfall Hg deposition should have a similar uncertainty as rainfall Hg deposition. Considering the possible interference for throughfall sample collection, the overall uncertainty of throughfall Hg deposition observation is estimated as ±20 %.

**4   Uncertainties in Hg deposition simulation**

**4.1   Uncertainties in models for Hg wet deposition**

### 4.1.1 Model for precipitation Hg wet deposition

Hg wet deposition through precipitation is an important process in global or regional chemical transport models (CTMs), such as GEOS-Chem and CMAQ-Hg (Lin et al., 2010; Y. Zhang et al., 2012; Bieser et al., 2014; J. Zhu et al., 2015; Horowitz et al., 2017). As shown in Eq. (1), precipitation Hg wet deposition is the product of the total Hg concentration in rainwater and the precipitation depth. The precipitation Hg concentration contains more uncertain factors. Hg in rainwater is mainly from the scavenging of GOM and PBM in both free troposphere and boundary layer. Based on the modeling work for Hg wet deposition in the United States using GEOS-Chem (Selin and Jacob, 2008), GOM and PBM contributed 89 % and 11 % to the total Hg wet deposition, respectively, and 60% of the GOM induced wet deposition originated from scavenging in the free troposphere. Seo et al. (2012) and Cheng et al. (2015) also reported higher scavenging coefficient for GOM than for PBM. Therefore, Hg redox chemistry in the free troposphere, aqueous phase Hg speciation, aqueous phase sorption, and the scavenging process tend to be the dominant sources of uncertainties (Lin et al., 2006; Lin et al., 2007; Cheng et al., 2015).

In the simulation of Hg wet deposition by the GEOS-Chem model, the uncertainty of precipitation depth is usually within ±10 % because it is based on assimilated meteorological observations from the Goddard Earth Observing System (GEOS) instead of meteorological models (Y. Zhang et al., 2012). Y. Zhang et al. (2012) conducted a nested-grid simulation of Hg over North America using GEOS-Chem, and reported the normalized bias of the annual Hg wet deposition flux to be ranging from −14 % to +27 % comparing to the MDN observations. Horowitz et al. (2017) used GEOS-Chem to reproduce observed Hg wet deposition fluxes over North America, Europe, and China and also got low bias (0–30 %). The CMAQ-Hg model exhibits a higher uncertainty level because the precipitation depth is simulated by meteorological models (e.g., MM5 or WRF) and its uncertainty has a strong impact on model prediction on Hg wet deposition (Lin et al., 2006). In the study of Bullock et al. (2009), the precipitation simulated by MM5 was averagely 12% greater than observed and the CMAQ simulation of Hg wet deposition was averagely about 15% above the MDN observations. However, different boundary conditions could cause a 25% difference (Bullock et al., 2009). Holloway et al. (2012) found that the CMAQ-Hg model underestimated wet deposition by 21 % on an annual basis and showed

average errors of 55 %. Based on the comparison between observed and modeled results and the sensitivity of key parameters, the overall uncertainty of precipitation Hg wet deposition simulation is estimated to be ±30 %.

**4.1.2 Model for non-precipitation Hg wet deposition**

Non-precipitation Hg wet deposition simulation has never been considered in CTMs, but performed in some individual studies with Hg concentration data for cloud, fog, dew or frost samples (Ritchie et al., 2006; Converse et al., 2014; Blackwell and Driscoll, 2015b). Non-precipitation deposition depth can be estimated using resistance models, analytical models or sophisticated atmosphere-soil-vegetation models. Katata (2014) reviewed different types of models for fog deposition estimation, and found the four most sensitive factors to be canopy homogeneity, droplet size spectra, droplet capture efficiency, and canopy structure. Since fog is the most important form of non-precipitation deposition, the overall uncertainty in the simulation of non-precipitation Hg wet deposition is estimated to be ±200 % or a factor of 3 based on the sensitivity analysis in the study of Katata (2014).

**4.2    Uncertainties in models for Hg dry deposition**

Hg dry deposition flux can be estimated by coupling speciated atmospheric Hg concentrations with dry deposition models (Wright et al., 2016). Therefore, in this part, the uncertainties of speciated Hg concentration measurements were first discussed, followed by the uncertainty analyses of Hg dry deposition models.

**4.2.1 Uncertainties in speciated Hg concentration measurements**

Although many new methods and apparatus have been or are being developed to better determine speciated Hg concentrations in ambient air, up to now the Tekran 2537/1130/1135 system is still the most widely used commercial instrument for continuous measurements of speciated Hg (Gustin et al., 2015). Regional and global monitoring networks such as Atmospheric Mercury Network (AMNet) and GMOS have all been using the Tekran systems and developed systematic quality assurance and quality control (QA/QC) protocols to assure data quality (Obrist et al., 2018). Therefore, this section is mainly to assess the uncertainties of the Tekran system.

Tekran 2537 uses a pair of gold trap cartridges (A/B) to capture GEM in order to achieve continuous observation and to reduce the uncertainty of GEM measurements. The standard operating procedure (SOP) of GMOS for the determination of GEM

requires the RPD of the average of five consecutive A trap concentrations and five
consecutive B trap concentrations to be less than 10 % (Sprovieri et al., 2017). In field
comparisons held by EMEP, the RSD from Tekran measurements are also generally
within 10 % (Aas, 2006). However, in the Reno Atmospheric Mercury
Intercomparison eXperiment (RAMIX) campaign, the RPD between two co-located
Tekran systems was as high as 25–35 % (Gustin et al., 2013). This was possibly
related to other factors, such as the configuration of the manifold, which could be
occasional or systemic. Therefore, considering the possible uncertainty brought by the
system setup, the overall uncertainty of GEM concentration measurements by the
Tekran system is estimated to be ±20 %.

Tekran 1130 uses a KCl-coated denuder to pre-concentrate GOM, and the collected

GOM is then thermally desorbed at 500 °C and converted to GEM for quantification.
A number of studies have reported the significant interference of ozone and humidity
on the GOM capture rate of the denuder (Lyman et al., 2010; Jaffe et al., 2014;
McClure et al., 2014; Gustin et al., 2015). McClure et al., (2014) found that the KCl-
coated denuder only captures 20–54 % $HgBr_2$ in the ambient air under the influence
of humidity and ozone. Huang et al. (2013) compared denuder- and membrane-based
methods, and reported that the KCl-coated denuder only captures 27–60 % of the
GOM measured by CEMs. Discrepancy with a factor of 2–3 at times was found
between the Tekran system and other new methods in the RAMIX campaign (Gustin
et al., 2013). Cheng and Zhang (2017) developed a numerical method to assess the
uncertainty of GOM measurements, and estimated the GOM concentrations measured
at 13 AMNet sites to be underestimated by a factor of 1.3 to more than 2. Gustin et al.
(2015) reported that the capture efficiency ratio of CEMs over the denuder method for
five major GOM compounds ranges from 1.6 to 12.6. Recent studies (Huang and
Gustin, 2015a; Huang et al., 2017) applied a correction factor of 3 for Tekran GOM
data when modeling dry deposition flux. Therefore, the overall uncertainty of the
GOM concentration measured by the Tekran system is estimated to be ±200 % or
within a factor of 3.

Tekran 1135 uses a quartz filter downstream the KCl denuder to collect $PM_{2.5}$, and

the collected fine particles are then thermally desorbed at 800 °C at a pyrolyzer and
converted to GEM for the quantification of PBM, or rather $PBM_{2.5}$. The uncertainties
in PBM concentration measurements have not been systemically assessed so far.
Gustin et al. (2015) pointed out that breakthrough of GOM from the upstream denuder
could result in the retention of GOM on the quartz filter and induce consequent PBM
overestimation. The RAMIX campaign showed that the RSD of PBM measurements
was 70–100 % when the Tekran systems were free standing (Gustin et al., 2013).
Coarse PBM is neglected in Tekran measurements with an impactor removing all
coarse particles. However, based on the estimation of Zhang et al. (2016b), about
30 % of PBM could be on coarse particles. Regarding the limited evidence from
previous studies, the overall uncertainty of the PBM concentration measured by the
Tekran system is estimated to be ±100 % or a factor of 2.
**4.2.2 Resistance model for GOM dry deposition**
Based on Eq. (4), the dry deposition velocity ($v_d$) is the key parameter in the
determination of Hg dry deposition flux. It can be estimated using a resistance model
(Zhang et al., 2002; Zhang et al., 2003):
$$v_d = \frac{1}{R_a + R_b + R_c} \tag{12}$$

where $R_a$ is the aerodynamic resistance depending on the meteorological conditions
and the land use category; $R_b$ is the quasi-laminar resistance, a function of friction
velocity and the molecular diffusivity of each chemical species (Zhang et al., 2002);
and $R_c$ is the canopy resistance which can be further parameterized as follows:
$$R_c = \left( \frac{1 - W_{st}}{R_{st} + R_m} + \frac{1}{R_{ns}} \right)^{-1} \tag{13}$$

where $W_{st}$ is the fraction of stomatal blocking under wet conditions; $R_{st}$ is the
stomatal resistance; $R_m$ is the mesophyll resistance; and $R_{ns}$ is the non-stomatal
resistance which is comprised of in-canopy, soil, and cuticle resistances. Cuticle and
soil resistances for GOM are scaled to those of $SO_2$ and $O_3$ by the following equation:
$$R_{x,\mathrm{GOM}} = \left( \frac{\alpha_{\mathrm{GOM}}}{R_{x,\mathrm{SO_2}}} + \frac{\beta_{\mathrm{GOM}}}{R_{x,\mathrm{O_3}}} \right)^{-1} \tag{14}$$

where $R_x$ is the cuticle or soil resistance; $\alpha$ and $\beta$ are two scaling parameters (Zhang
et al., 2003; L. Zhang et al., 2012). Among the numerous parameters in the resistance
model the two scaling factors for the non-stomatal resistance components regarding
the solubility and reactivity of the chemical species are the most sensitive ones. The
values for $HNO_3$ ($\alpha=\beta=10$) used to be applied in the model for GOM (Marsik et al.,
2007; Castro et al., 2012; L. Zhang et al., 2012). However, some other studies found
the values for HONO ($\alpha=\beta=2$) are probably more suitable for GOM due to equivalent

effective Henry's Law constants ($H^*$) between HONO and $HgCl_2$ (Lyman et al., 2007). Huang and Gustin (2015a) indicated that no single value could be used to calculate GOM dry deposition due to the unknown GOM compounds. Various values for the two scaling parameters ($\alpha=\beta=2$, 5, 7 and 10) were used in Huang et al. (2017) to identify dominant GOM deposition species.

The uncertainties of $R_a$ and $R_b$ are estimated to be generally small, within the range of ±30 % (Zhang et al., 2003; Huang et al., 2012a), while the uncertainty of $R_c$ usually has a larger impact, especially through the selection of $\alpha$ and $\beta$. Lyman et al. (2007) changed the values of $\alpha$ and $\beta$ from 2 to 10, and found a 120% enhancement of $v_d$. With a correction factor of 3 for the GOM concentration measured by Tekran, Huang and Gustin (2015a) got similar modeled and measured GOM dry deposition values with bias of up to ±100 %. Huang et al. (2017) also applied the correction factor of 3, tested different values of $\alpha$ and $\beta$, and found the bias of GOM dry deposition simulation to be up to a factor of 2.5. As discussed above, the overall uncertainty of the GOM concentration measured by Tekran is within a factor of 3. If the GOM dry deposition simulation is directly based on the Tekran GOM data, its uncertainty level would be much higher than a factor of 3. However, recent studies (Huang et al., 2014; Huang and Gustin, 2015a; Huang et al., 2017) have used a correction factor of 3 for GOM concentration data which offsets the uncertainty of GOM dry deposition. Therefore, the overall uncertainty in GOM dry deposition simulation is estimated to be a factor of 2.5 or ±150 %.

**4.2.3 Resistance model for PBM dry deposition**

For PBM dry deposition, resistance models regarding both fine and coarse particles are more and more widely applied based on the theory that $v_d$ for atmospheric particles strongly depend on particle size (Dastoor and Larocque, 2004; Zhang et al., 2009; Zhang and He, 2014). Many independent studies (Fang et al., 2012b; Zhu et al., 2014) showed that Hg in coarse particles constitutes a large mass fraction of the total PBM, which was previously neglected. PBM measured by Tekran 2537/1130/1135 only considers fine particles. Based on measurements of particle size distributions and Hg mass distribution between fine and coarse particles, Zhang et al. (2016b) assumed that coarse particles account for 30 % of the total PM, and the Hg mass concentrations on fine and coarse particles are consistent. Taking coarse particles into consideration, the total PBM dry deposition can be calculated as follows (Zhang et al., 2016b):

$$F_{\mathrm{dry,PBM}} = C_f \left( v_f + \frac{f}{1-f} v_c \right) \tag{15}$$

where $F_{\mathrm{dry,PBM}}$ is the total PBM dry deposition flux; $C_f$ is the mass concentration of
PBM in fine particles; $v_f$ and $v_c$ are the dry deposition velocities of PBM for fine and
coarse particles, respectively; and $f$ is the mass fraction of PBM in coarse particles. $v_f$
and $v_c$ can be calculated using the following equation (Zhang et al., 2001):
$$v_x = v_g + \frac{1}{R_a + R_s} \tag{16}$$

where $v_x$ is $v_f$ or $v_c$; $v_g$ is the gravitational settling velocity; $R_a$ is the aerodynamic
resistance; and $R_s$ is the surface resistance which can be parameterized as a function of
collection efficiencies from Brownian diffusion, impaction, and interception
mechanisms (L. Zhang et al., 2012; Zhang et al., 2016b). Zhang and He (2014) have
developed an easier bulk algorithm based on the $v_x$ scheme of Zhang et al. (2001) to
make this model more widely applicable in monitoring networks.
Zhang et al. (2001) conducted a model comparison with two PBM dry deposition
schemes, and the results showed that the differences between models are generally
within the range of 20 %. However, recent studies found the proportion of coarse
particles plays a crucial role in the evaluation of PBM dry deposition velocity (Zhang
et al., 2016b). Zhang et al. (2016b) assumed that 30 % of the total PBM mass is on
coarse particles, and found that 44 % PBM deposition was caused by coarse particle
deposition. We tested the model used by Zhang et al. (2016b), and found a 2-fold
change when we increased the coarse PBM proportion from 30 % to 50%. In other
words, the uncertainty of the PBM deposition velocity could be as high as ±100 %. As
discussed above, the overall uncertainty of the PBM concentration measured by
Tekran is about ±100 %. Considering both aspects and applying the calculation
method based on Eq. (2), the overall PBM uncertainty in GOM dry deposition
simulation is estimated to be ±150 %.
**4.2.4 Bidirectional model for GEM dry deposition**
GEM dry deposition can also be calculated using the resistance model with different
parameters. However, the re-emission and natural emission of GEM must be taken
into consideration. Net GEM dry deposition is estimated from the difference between
the estimated unidirectional deposition flux and the modeled total re-emission plus
natural emission in the resistance model (L. Zhang et al., 2012).
A bidirectional air-surface exchange model modified from the resistance model is
more and more recommended in recent years (Zhang et al., 2009; Bash, 2010; Wang
et al., 2014; Zhang et al., 2016b; Zhu et al., 2016). In the bidirectional scheme, the
GEM dry deposition flux can be calculated as follows (Zhang et al., 2009):
$$F_{\text{dry,GEM}} = \frac{\chi_a - \chi_c}{R_a + R_b} \qquad (17)$$

$$\chi_c = \left( \frac{\chi_a}{R_a + R_b} + \frac{\chi_{st}}{R_{st} + R_m} + \frac{\chi_g}{R_{ac} + R_g} \right) \left( \frac{1}{R_a + R_b} + \frac{1}{R_{st} + R_m} + \frac{1}{R_{ac} + R_g} + \frac{1}{R_{cut}} \right)^{-1} \quad (18)$$

where $F_{\text{dry,GEM}}$ is the net GEM dry deposition flux; $\chi_a$ is the GEM concentration at a
reference height; $R_a$, $R_b$, $R_{st}$, $R_m$, $R_{ac}$, $R_g$ and $R_{cut}$ are aerodynamic, quasi-laminar,
stomatal, mesophyll, in-canopy aerodynamic, ground surface and cuticle resistances,
respectively (Zhang et al., 2016b); and $\chi_{st}$ and $\chi_g$ are canopy, stomatal and ground
surface compensation points, respectively. Based on observations on different land use
categories, Wright and Zhang (2015) have proposed a range of $\chi_{st}$ and $\chi_g$.
The studies of L.Zhang et al. (2012) and Zhang et al. (2016b) have shown the great
importance of the previously neglected GEM dry deposition. Due to the presence of
natural and re-emission of GEM, the net GEM dry deposition has a higher uncertainty
level than GOM and PBM dry deposition. Although both the studies of L. Zhang et al.
(2012) and Zhang et al. (2016b) reported the uncertainty of net GEM dry deposition to
be averagely about a factor of 2, there were many exceptions (over a factor of 2–5)
according to L. Zhang et al. (2012), especially when the net GEM dry deposition
fluxes were at low level. Based on the above concern and the sensitivity analysis
conducted in the study of Zhang et al. (2016b), the overall uncertainty of the net GEM
dry deposition simulation is within a factor of 2 or ±100 % when GEM dominates the
total Hg dry deposition, while it could be as high as a factor of 5 or ±400 % when
GOM+PBM dominate the total dry deposition. According to this estimation, the
overall uncertainty of the total dry deposition is in the range of ±(100–150) %. It tends
to increase when the dominance of dry deposition shifts from GEM to GOM+PBM.
**4.3   Uncertainties in models for forest Hg deposition**
The study of Wang et al. (2016a) is to date the only modeling study for litterfall Hg
deposition. Monte Carlo simulation was adopted to assess the global Hg deposition
through litterfall based on the measured litterfall Hg concentrations and the global
litterfall biomass distribution. The estimated global annual Hg deposition through
litterfall was reported to be 1180 t with a relative uncertainty of ±60 %. There is no
modeling study on throughfall Hg deposition so far. Consequently, we can only use
the overall uncertainty of wet and dry deposition simulation to represent throughfall,
which will be discussed in the next section.

## 5 Summary of uncertainties in Hg deposition to terrestrial surfaces

Based on the review work above, the overall uncertainties of wet, dry, and forest Hg
deposition can be calculated using the following equation:
$$\delta_{A+B} = \frac{U_{A+B}}{F_{A+B}} = \frac{\sqrt{U_A^2 + U_B^2}}{F_{A+B}} = \frac{\sqrt{F_{A+B}^2 P_A^2 \delta_A^2 + F_{A+B}^2 P_B^2 \delta_B^2}}{F_{A+B}} = \sqrt{P_A^2 \delta_A^2 + P_B^2 \delta_B^2} \qquad (19)$$
where $\delta_A$, $\delta_B$, and $\delta_{A+B}$ are the relative uncertainties of Part $A$, Part $B$, and the total
deposition flux, respectively; $U_A$, $U_B$, and $U_{A+B}$ are the absolute uncertainties of them,
respectively; $F_{A+B}$ is the total deposition flux; and $P_A$ and $P_B$ are the proportions of
Part $A$ and Part $B$ deposition fluxes, respectively.
Table 1 summarizes the previously estimated relative uncertainties for wet, dry, and
forest Hg deposition fluxes. Although the uncertainty of precipitation Hg deposition
flux is low (±12 % and ±30 % for observation and simulation, respectively), the
uncertainty of non-precipitation Hg deposition has been neglected. Due to the
condensation effect, non-precipitation deposition could contribute equivalent or even
larger proportion to Hg wet deposition than rainfall (Stankwitz et al., 2012; Blackwell
and Driscoll, 2015b; Weiss-Penzias et al., 2016b; Gerson et al., 2017). Considering
the global area of hotspot regions for cloud, fog, dew, and frost, such as alpine and
coastal regions, the overall contribution of non-precipitation deposition to Hg wet
deposition is approximately 5–10 %. Given the high uncertainty level of non-
precipitation Hg deposition, the overall uncertainties in the observation and simulation
of global Hg wet deposition are estimated to be ±(20–30) % and ±(30–35) %,
respectively.
Hg dry deposition has a much larger uncertainty level than wet deposition from
both observation and simulation perspectives. High GOM deposition fluxes were
exhibited in North America, while high PBM deposition fluxes were found in East
Asia (Wright et al., 2016). Based on the global observation and simulation data
(Wright et al., 2016; Zhang et al., 2016b), the ratio of global GOM dry deposition
over PBM dry deposition could be in the range of 1:1 to 3:1, and the ratio of global
GEM dry deposition over RM (GOM+PBM) dry deposition could be in the range of
1:9 to 9:1. Therefore, the overall uncertainties in the observation and simulation of
global Hg dry deposition are estimated to be ±(55–90) % and ±(90–130) %,
respectively.
Without studies specifically on throughfall deposition modeling, the uncertainty of
throughfall Hg deposition simulation has been estimated based on the uncertainties of
both wet and dry deposition simulation, and turned out to be up to ±90 %. Studies on
both litterfall and throughfall Hg deposition (Larssen et al., 2008; Navrátil et al.,
2014; Luo et al., 2016; Ma et al., 2015; Fu et al., 2016a; Wang et al., 2016a; Gerson et
al., 2017) showed that the relative contributions of litterfall and throughfall could be
in the range of 2:3 to 4:1. Accordingly, the overall uncertainties in the observation and
simulation of global forest Hg deposition are estimated to be ±(20–25) % and ±(50–
60) %, respectively.
Based on global and regional modeling studies (Selin and Jacob, 2008; Wang et al.,
2016a; UN Environment, 2019), the relative contributions of wet, dry, and litterfall
Hg deposition are estimated to be approximately 1:2:1. With the previously estimated
uncertainty ranges for wet, dry, and litterfall deposition, the overall uncertainties in
the observation and simulation of global total Hg deposition are calculated to be
±(30–50) % and ±(50–70) %, respectively. It should be noted that the low overall
uncertainty for observation can only be achieved when Hg deposition networks are
established worldwide.
**6    Implications and future research needs**
With a big effort of literature review, this study has estimated the uncertainties in the
observation and simulation of global Hg deposition to terrestrial surfaces through
different pathways. The implications from the comprehensive uncertainty analysis and
the derivative research needs in the future are as follows:
(1) The observation methods for both wet and forest Hg deposition fluxes have low
uncertainty levels. Although large uncertainties still exist in the methods for Hg dry
deposition measurements, the overall uncertainty in global Hg deposition observation
can be as low as ±(30–50) % as long as global dry deposition monitoring networks for
GOM, PBM and GEM are established. Optimized surrogate surfaces and DFCs are
economic approaches for RM and GEM measurements, respectively, and could be
recommended for the global dry deposition network.

(2) Methods with high time resolution for the accurate measurements of GOM and PBM concentrations are in urgent needs. The KCl denuder-based method for GOM measurements has significant underestimation. The application of a correction factor of 3 could reduce the uncertainty in GOM dry deposition simulation. However, this correction factor is not universally applicable. Different humidity levels or ozone concentrations lead to a significant change in underestimation. Different chemical forms of GOM (e.g., $HgCl_2$, $HgBr_2$, $HgO$, $HgSO_4$, etc.) also have different KCl capture efficiencies. On account of the GOM dry deposition velocity, the chemical form of GOM also plays a crucial role. Different model parameterizations should be applied for different GOM species. Therefore, quantification methods for measuring different GOM species need to be developed to improve the simulation of GOM dry deposition flux.

(3) The contribution of GEM dry deposition to the total global Hg deposition is still unclear, which leads to the extremely large uncertainty in GEM dry deposition simulation. More comparisons between observation and simulation of the GEM dry deposition flux should be conducted to improve model parameterization. Moreover, the GEM deposition process is complicated in forests. It is useful to measure the above-canopy apparent deposition flux, the under-canopy dry deposition flux, the litterfall deposition flux, and the throughfall deposition flux at the same site to get a more comprehensive understanding of the process.

(4) Non-precipitation Hg wet deposition has been neglected in the global monitoring networks and modeling studies. Cloud, fog, or even dew and frost Hg deposition could be quite important in hotspot regions, such as alpine and coastal areas. It could be enriched in aqueous Hg and affect other deposition processes, or in other words, change the overall Hg residence time. Extremely large uncertainties still exist in both observation and simulation of non-precipitation Hg wet deposition. More standardized sampling methods are required for long-term observation of non-precipitation Hg wet deposition.

(5) Asia has the highest atmospheric Hg concentration level. However, the Hg deposition studies in Asia are still quite limited. Hg wet deposition network has not been established in Asia, and there are only a few scattered studies on dry deposition in East Asia. The Hg wet and dry deposition processes in Asia could be quite different from those in North America and Europe because of the high atmospheric Hg and high PM condition in Asia.

*Author contribution.* Dr. Lei Zhang designed the review framework. Dr. Lei Zhang and Peisheng Zhou did the most literature review work with contributions from Shuzhen Cao and Dr. Yu Zhao. Dr. Lei Zhang prepared the manuscript with contributions from all co-authors.

*Acknowledgements.* This review work was supported by the National Natural Science Foundation of China (No. 21876077) and the Fundamental Research Funds for the Central Universities (No. 14380080, No. 14380092, and No. 14380124).

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

 **Table Captions**

 **Table 1.** Summary of relative uncertainties of different types of Hg deposition to

 terrestrial surfaces.

 **Table 1.** Summary of relative uncertainties of different types of Hg deposition to
 terrestrial surfaces.

| Type of Hg deposition | Relative uncertainty in observation (%) | Relative uncertainty in simulation (%) |
|---|---|---|
| **Wet deposition** | ±(20–30) | ±(30–35) |
| Precipitation | ±12 | ±30 |
| Cloud, fog, dew, and frost | ±300 | ±200 |
| **Dry deposition** | ±(55–90) | ±(90–130) |
| GOM dry deposition | ±70 | ±150 |
| PBM dry deposition | ±100 | ±150 |
| GEM dry deposition | ±100 | ±100 (GEM dominates) ±400 (RM dominates) |
| **Forest deposition** | ±(20–25) | ±(50–60) |
| Litterfall | ±30 | ±60 |
| Throughfall | ±20 | ±90 |
| **Overall** | ±(30–50) | ±(50–70) |

**Figure Captions**

**Figure 1.** Global distribution of the observed Hg wet deposition fluxes by observation networks around the world ($\mu g\ m^{-2}\ yr^{-1}$).

**Figure 2.** Hg wet deposition fluxes (cyan columns with black bars as standard deviations) and annual precipitation (orange dots) for different terrestrial surface types. "Water" stands for the terrestrial surfaces near water. The numbers in brackets stand for the numbers of samples.

**Figure 3.** Global distribution of the (a) GOM, (b) PBM, and (c) GEM dry deposition fluxes ($\mu g\ m^{-2}\ yr^{-1}$) from observation-based estimation.

**Figure 4.** Relationship between the elevation and the GOM dry deposition flux.

**Figure 5.** Comparison between the GOM dry deposition fluxes from direct observations and from model simulations based on measurements of GOM concentrations. The numbers in brackets stand for the numbers of samples.

**Figure 6.** Dry deposition fluxes (cyan columns with black bars as standard deviations) of (a) GOM, (b) PBM and (c) GEM for different terrestrial surface types. "Water" stands for the terrestrial surfaces near water. The numbers in brackets stand for the numbers of samples.

**Figure 7.** Litterfall Hg deposition fluxes (cyan columns with black bars as standard deviations) and Hg concentrations in litterfall (orange dots) for different terrestrial surface types. The numbers in brackets stand for the numbers of samples. DB stands for deciduous broadleaf forests, DN stands for deciduous needle leaf forests, EB stands for evergreen broadleaf forests, and EN stands for evergreen needle leaf forests.

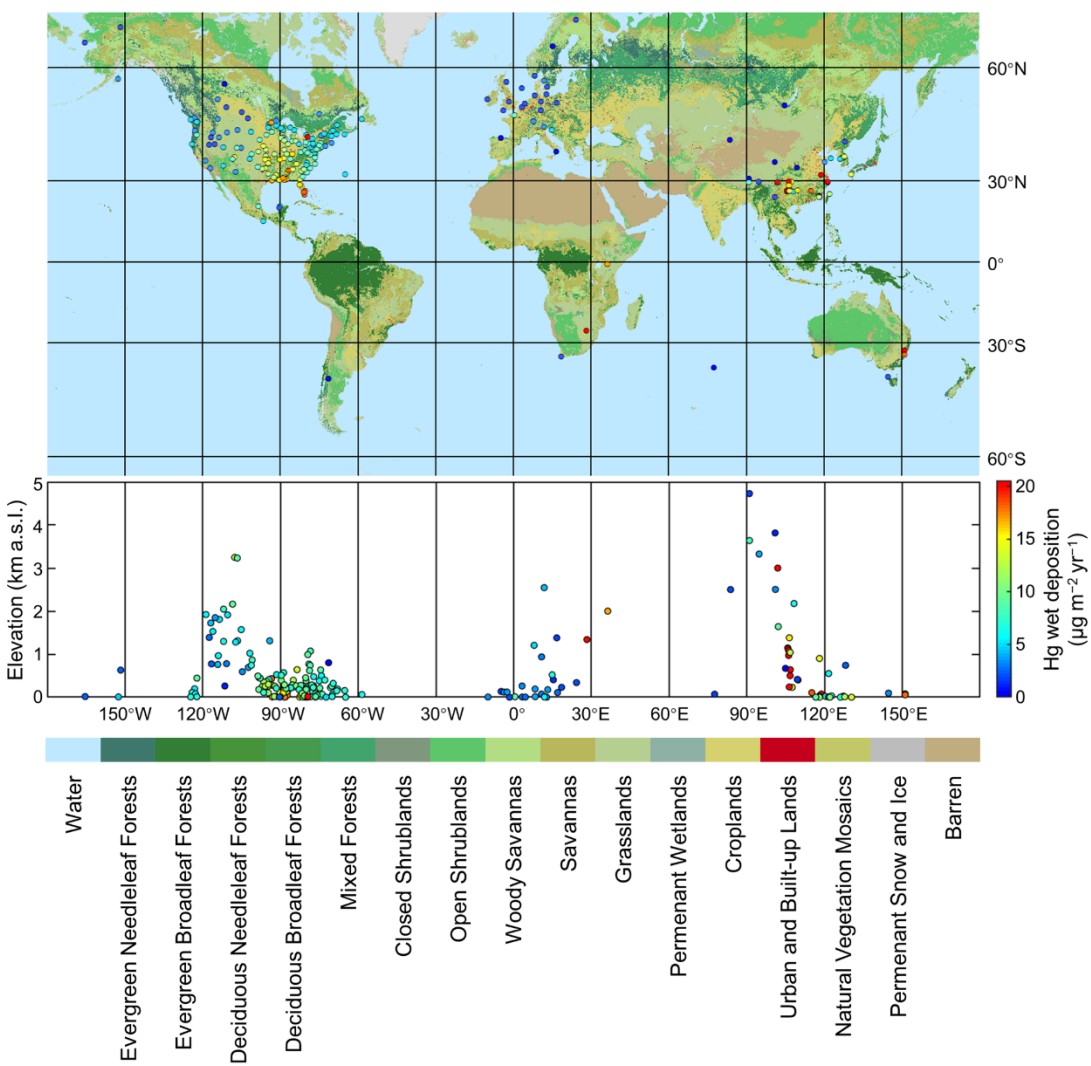


**Figure 1.** Global distribution of the observed Hg wet deposition fluxes by observation

networks around the world ($\mu g\ m^{-2}\ yr^{-1}$).


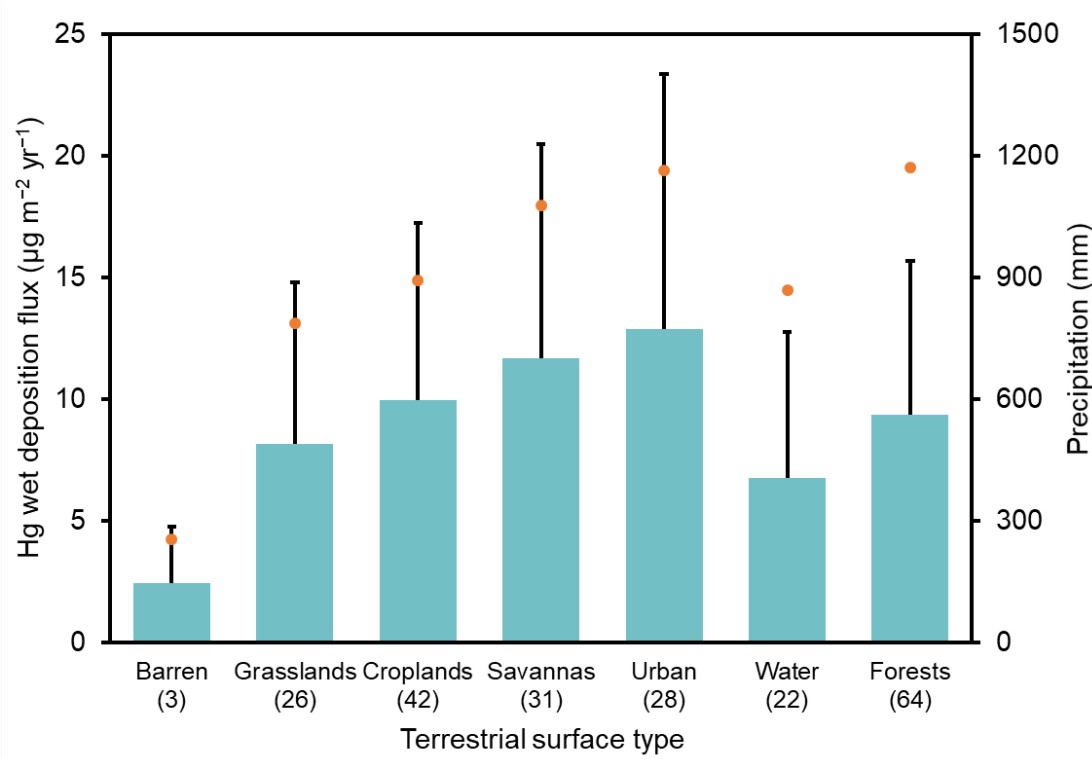


**Figure 2.** Hg wet deposition fluxes (cyan columns with black bars as standard
deviations) and annual precipitation (orange dots) for different terrestrial surface
types. "Water" stands for the terrestrial surfaces near water. The numbers in brackets
stand for the numbers of samples.

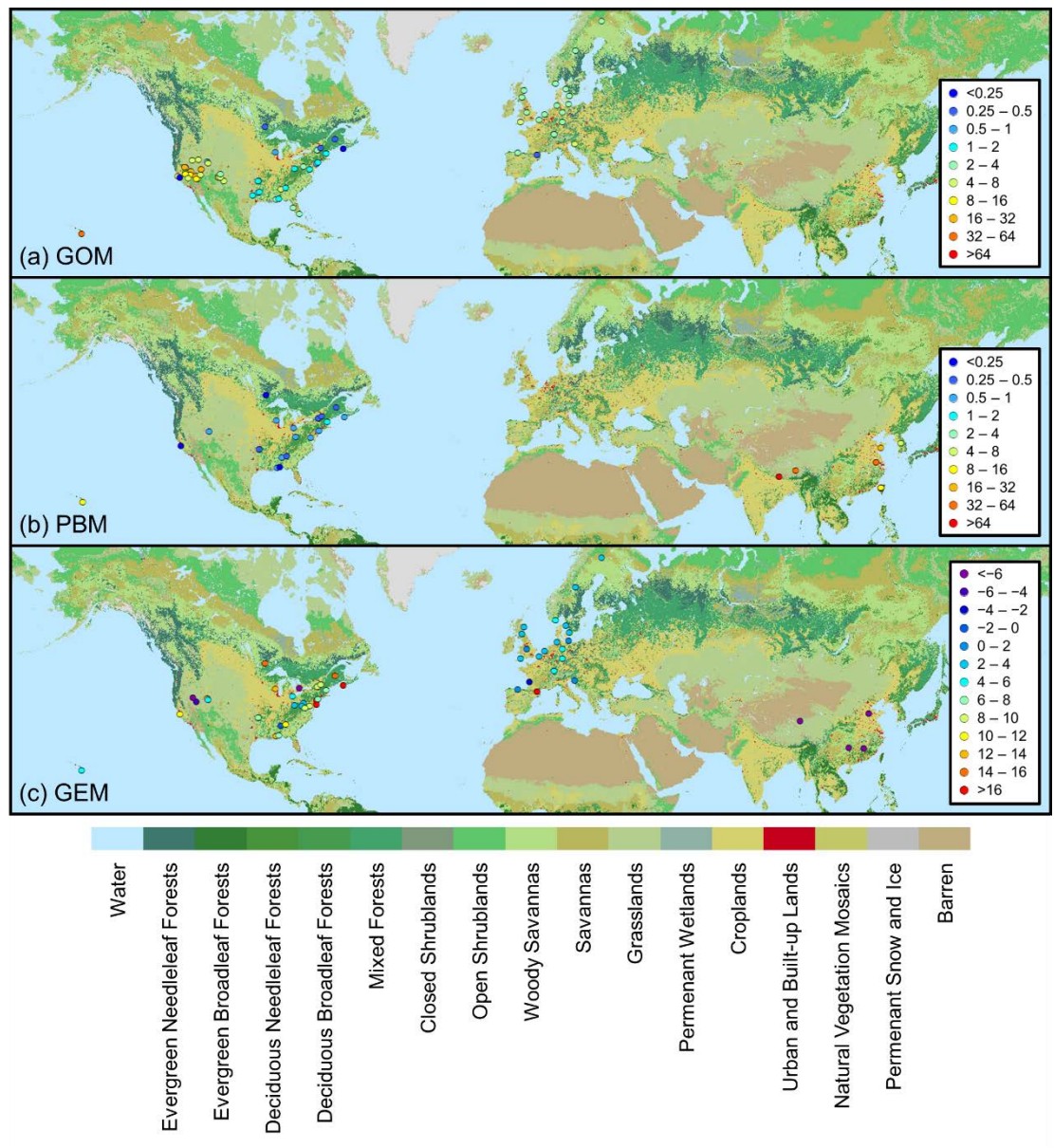

**Figure 3.** Global distribution of the (a) GOM, (b) PBM, and (c) GEM dry deposition fluxes ($\mu$g m$^{-2}$ yr$^{-1}$) from observation-based estimation.

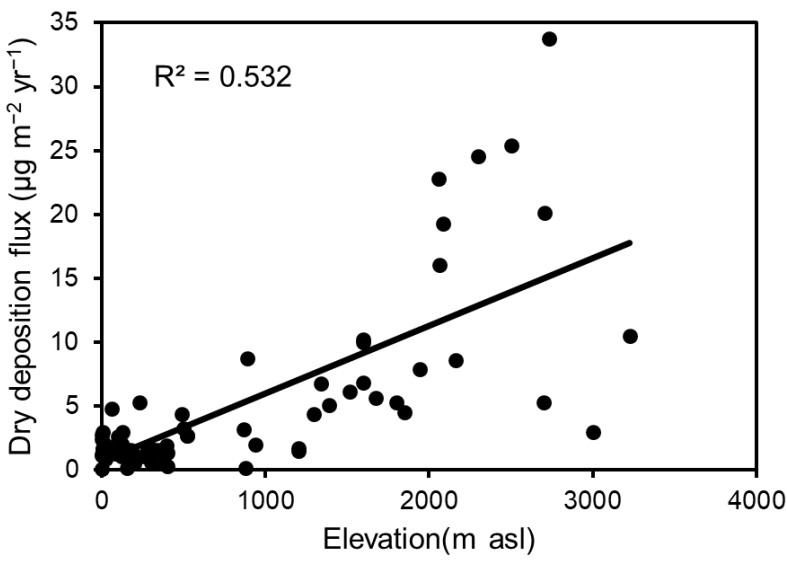

**Figure 4.** Relationship between the elevation and the GOM dry deposition flux.

1604

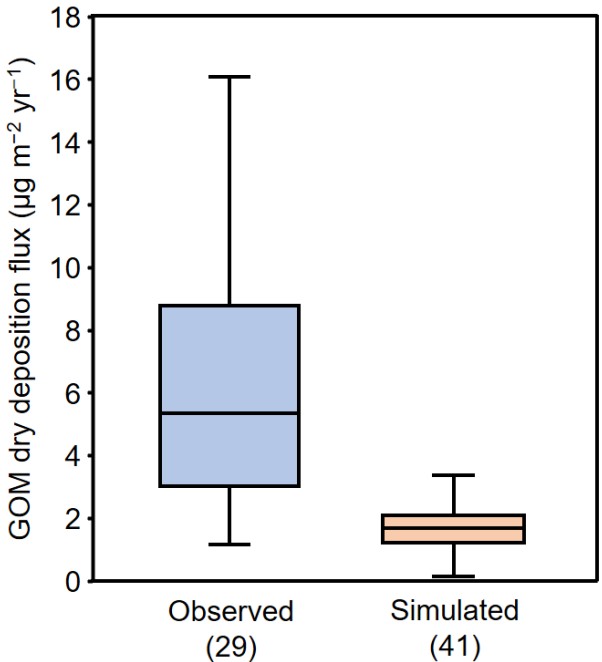

**Figure 5.** Comparison between the GOM dry deposition fluxes from direct observations and from model simulations based on measurements of GOM concentrations. The numbers in brackets stand for the numbers of samples.

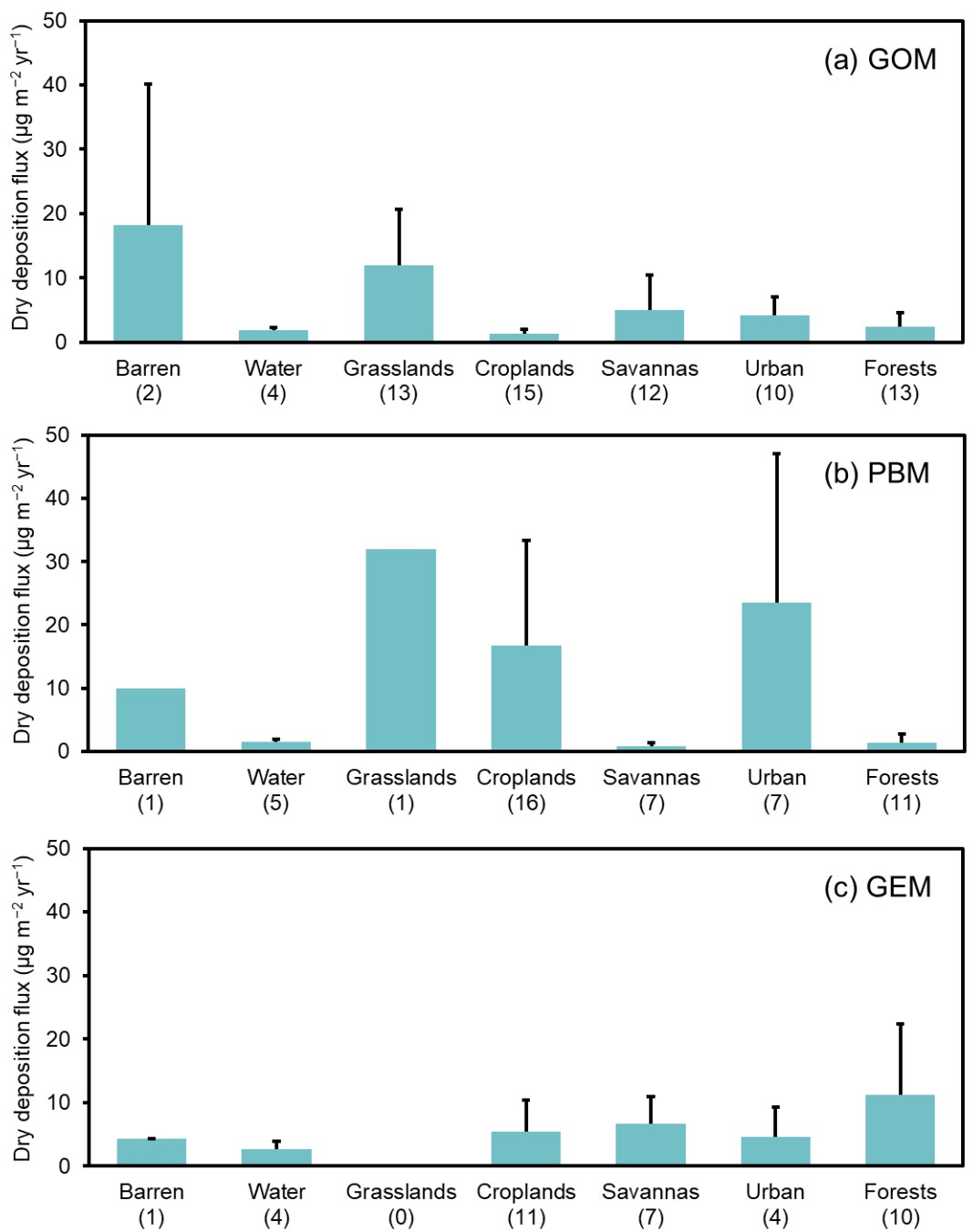

1610

**Figure 6.** Dry deposition fluxes (cyan columns with black bars as standard deviations)
of (a) GOM, (b) PBM and (c) GEM for different terrestrial surface types. "Water"
stands for the terrestrial surfaces near water. The numbers in brackets stand for the
numbers of samples.

1615

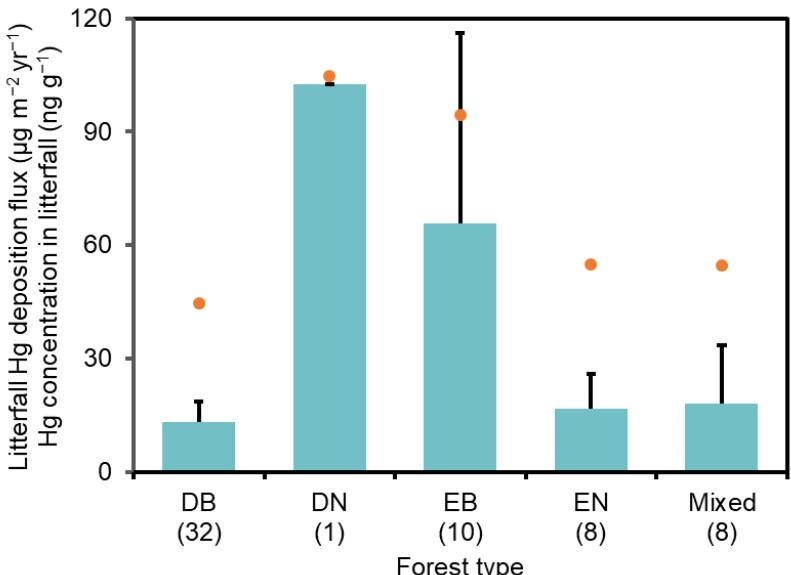

1616

**Figure 7.** Litterfall Hg deposition fluxes (cyan columns with black bars as standard deviations) and Hg concentrations in litterfall (orange dots) for different terrestrial surface types. The numbers in brackets stand for the numbers of samples. DB stands for deciduous broadleaf forests, DN stands for deciduous needle leaf forests, EB stands for evergreen broadleaf forests, and EN stands for evergreen needle leaf forests.