# Peer review of "Uncertainties in the observation and simulation of"

_Atmospheric Chemistry and Physics, 2019_

## Referee Comment (RC1) · Anonymous Referee #3 · 31 May 2019

Overall comments:

This is a review paper discussing global Hg deposition. This discussion is very important for Hg research community; however, the authors are not doing a very good job to deliver key messages from the review process. A review paper is to summarize findings from previous related studies and provide approaches/methods/concepts to help the research community moving forward. However, I did not see the authors clearly made all these points in their article.

In general, this paper is not easy to follow, the authors jump from one topic to another. They did not do advanced discussion. In more paragraphs, they only described methods and data, and probably two/three sentences to summarize/discuss what they learn from these methods/data. There is nothing inspiring readers. A review paper should

do better than that.

Their conclusions/summaries are not new. Gustin's group has published couple review articles discussing the first three aspects in 2015, and the 4th aspect has been mentioned in multiple previous articles. I really do not find any new concepts in this article, and how can we solve the difficulties that the Hg research community is facing. For example, do the authors have any suggestion to understand behaviors of various GOM compounds in the atmosphere?

I agree this is an important research field and there are gaps which make scientists cannot fully understand global Hg cycle. A review paper related to this topic should be published to draw attention from environmental research groups. However, the way that this paper is done cannot provide useful information to scientists.

I suggest the authors re-think about the article structure and put more efforts on advanced discussions.

Specific comments: Abstract is read more like a summary than an abstract. I suggest to re-write the abstract and focus on your key aspects. Moreover, the authors must provide some potential solutions/suggests for each gap that are discussed in their conclusions.

Introduction is fine, but this is a review paper. There are more previous Hg review articles, such as Selin et al., 2007, and some key finding paper are not included in this review paper, such as Moore et al., 2014 Nature. These articles might not be directly linked to Hg deposition, but they do have indirect impacts on Hg deposition. After reading this article, I feel the authors focus on the measuring methods and numeric models, but do not discuss in advance about global deposition processes.

Methods section: A summary table or multiple summary tables would help the readers to read through this section.

- Surrogate surface: the key point of this method is the surface affinity and fluent conditions near surface, but I did not see the authors discuss these here. Huang et al., 2011 published a paper discussing fluent conditions near KSS surface, and how this impacts mass transfer.

- Enclosure methods: Choi and Holsen 2008/2009 articles are also important, and the authors did not discuss about the bio-process/photo-process related to Hg reduction in DFC.

- Micrometeorological methods: This method has been used to understand GOM flux as well, but no discussion here.

- In forests: Choi and Holsen 2009, and there are more articles from Driscoll's group discussing Hg cycle in forests.

GOM resistance: page 10 line 299-310, Gustin et al., 2015 has summarized this, this is not a new idea. I just feel, the authors are writing a review article, but they are repeating the concepts from the summaries in other's review articles without adding their new thoughts.

Page 13 line 401-402, is ambient concentrations not important?

Line 404-405, figure 2 indicates ambient concentrations could be important.

Page 14, line 412-414, Europe has . . .. . ..., any ambient data to support this argument?

Line 427, deposition fluxes concentrations, what does "fluxes concentrations" mean?

Line 435-439, the authors should explain why they are showing significantly different? Different surface affinity?

Page 17, line 537-540, different surface (eg forest vs grassland), there are many differences between these two surface types, such as leaf area index, but the authors just simply summarized all these difference depositions based on chemistry and not talking about the characteristic of surfaces.

---

## Referee Comment (RC2) · Anonymous Referee #2 · 6 Jun 2019

This manuscript presents a review of speciated atmospheric mercury (Hg) deposition to the terrestrial surfaces on a globe scale. The topic is relevant to the Atmospheric Chemistry and Physics. However, the scientific contribution could be enhanced by a more inclusive review and a more depth discussion that highlight the advancement, challenges, and directions for future research. The presentation could be improved as well. My specific comments and suggestions are listed below.

Major concerns

1. A method section is missing. The authors may want to provide a Methodology section to cover the following items, how the literature search/review was conducted, what is the scope of the literature search, what are the primary source of publications (e.g. peer reviewed journal articles, government reports), restrictions if any (e.g. by

year of publication, or by language).

2. The scope of the review needs more justification. The title reads, "Global deposition of speciated atmospheric mercury to terrestrial surfaces: an overview". The rational of excluding the water surfaces (Figures 1, 6, 7 do include water through) and snow/ice over land should be presented.

3. The scientific contribution could be enhanced significantly. The manuscript as written is a somewhat descriptive presentation of estimation methods (sections 2 and 3) and Hg deposition values (sections 4 and 5). Consequently, there is a lack of new insights and findings. The authors are encouraged to conduct a rigorous research leading to more depth discussion that highlights the advancement, challenges, and directions for future research. Some potential topics are listed below (also see sample papers and a sample weblink at the end) 1) Comparison of co-located measurements with different techniques 2) Comparison of Hg deposition estimates by different models 3) Model-measurement comparison 4) Observed/predicted changes in Hg deposition due to changes in quantity of Hg emissions in local, regional or globe scale 5) Observed/predicted changes in Hg deposition due to changes in profiles (e.g. the percentage of each Hg species in total emission) of Hg emissions in local, regional or globe scale 6) Contributions to observed/simulated Hg dry deposition from different sources or regions 7) The major sources of uncertainty in Hg deposition estimates and how to reduce those uncertainties 8) What is the knowledge or data gap (relevant to Hg deposition) that hinders our understand of the global Hg cycle, or the development and evaluation of emission control measures?

4. The "Bidirectional air-surface exchange model for GEM" is presented. However, dry deposition of GEM is estimated in many field studies and model simulations, including most GEM dry deposition data presented in the manuscript. Thus, the authors may want to include dry deposition models of GEM.

5. Please provide facts to support your statements, e.g. "For PBM dry deposition, a

size-segregated resistance model is more and more widely applied" (L312)

Presentation

1. Most materials presented in sections 2 and 3 can be found in previous review/research papers, because those techniques have been around for a while. The authors could provide a summary table and direct the interested readers to those review/research papers, instead of a lengthy description of each method. Another option is to provide a comparative review of those methods and to include strength, weakness, recent advancements if any, and application issues.

2. Section 4.3 (Forest deposition or Deposition over forests) could be better placed in section 5 (Global Hg deposition on different terrestrial surfaces).

3. If the authors decided to keep the equations, please 1) provide unit of each variable, 2) provide the source of each equation, 3) clarify the expansion factor in equations (8) and (9). Is it an expansion from a measurement in a small area to a forest? 4) explain how to calculate two resistances with equation (16).

4. Please state the mechanism of Hg deposition via cloud/fog at high elevation sites (L258).

5. L456, the authors many want to distinguish the net emission fluxes from "natural GEM emission sources".

6. Figure 6, "precipitation levels" or "annual precipitation"?

7. The papers from which data were obtained to generate each figure could be tabulated and presented as Supplement Information.

Editorial suggestions

The use of English language is largely satisfactory. However, there is much room of improvement. Some examples are listed below.

1. There are quite a few awkward sentences and word choices, e.g. "Ci is the total Hg concentration in precipitation water" (L193), "Usually, GOM and PBM contribute equivalently to Hg wet deposition (Cheng et al., 2015)." (L206), add "GEM dry deposition is equivalent to GOM and PBM dry deposition, even significantly higher than in forests" (L535), "consequently exhibit significantly high litterfall Hg deposition fluxes." (L560), "Water surfaces could affect Hg wet deposition through fog scavenging." (L580), "The contribution GEM dry deposition has been underestimated previously." (L596), "Cloud, fog or even dew Hg deposition needs careful investigation" (L599), please rephrase.

2. There are some contradicting or confusing statements, e.g. "Based on available measurements of PBM size distributions and fine/coarse PBM mass ratios, Zhang et al. (2016b) assumed 30% of the total PBM mass to be coarse particles in order to estimate total PBM dry deposition flux based on the theory that PBM has the same proportion in both fine and course particles." (L318)

3. Avoid the use of first person, i.e. "we".

Sample model-model and model-measurement comparison papers:

Holmes, H. A., E. R. Pardyjak, K. D. Perry, and M. L. Abbott, 2011. Gaseous dry deposition of atmospheric mercury: A comparison of two surface resistance models for deposition to semiarid vegetation, J. Geophys. Res., 116, D14306, doi:10.1029/2010JD015182.

Wright, L.P. and L. Zhang, 2015. An approach estimating bidirectional air-surface exchange for gaseous elemental mercury at AMNet sites. J. Adv. Model. Earth Syst., 7, 35–49. (L1088)

Ye, Z., Mao, H., Driscoll, C.T., Wang, Y., Zhang, Y., Jaeglé, L. (2018) Evaluation of CMAQ Coupled With a State-of-the-Art Mercury Chemical Mechanism (CMAQ-newHg-Br). Journal of Advances in Modeling Earth Systems 10, 668–690. 588. https://doi.org/10.1002/2017MS001161

[Figure]

Sample weblink to find more papers:

https://www.researchgate.net/profile/Shuxiao_Wang

---

## Referee Comment (RC3) · Anonymous Referee #1 · 10 Jun 2019

This paper summarizes the important processes controlling atmospheric deposition of Hg. The topic is important, and new knowledge is available in the literature, so a review paper on this topic is a good and useful product to the broad scientific research community. However, there have been recent review papers that have largely covered the same topics and ideas, which leaves some doubt about this paper as one that makes a large contribution to the literature. For example, the abstract does not put forth many new ideas. There are a few missed opportunities such as when cloud/fog scavenging is mentioned the authors state: "the influence of cloud/fog scavenging is easy to neglect". The authors should be more quantitative in their language so as to provide scientists with more concrete information on relationships and processes. Another example in the abstract that is a missed opportunity to provide some detailed information is the

last line: "Future research needs have been proposed based on the current knowledge of global mercury deposition to terrestrial surfaces". This statement is too vague and does not provide much substance. For example, in the conclusion, the 4th recommendation regarding fog, cloud, and dew is The field "requires more standardized sampling methods". This is too vague and does not translate into a roadmap for improving the science. My suggestion is that the authors rethink their main focus of this paper – maybe all of deposition is too broad – and provide more insights and proscriptions for future research and/or data gaps. The authors have cited a large number of references and have done considerable research in the field. An improved focus would sharpen the discussion and make the paper more interesting to read.

One minor comment I have is that the following statement does not make sense to me: "The slope of the relationship implies the Hg concentration in precipitation. Europe has the flattest slope among all regions, indicating its lowest Hg pollution level around the world." Europe has the lowest Hg pollution level around the world? That does not seem correct.

---

## Author Comment (AC2) · 19 Aug 2019

Here are our responses to comments from Reviewer #1. Please find the revised manuscript in Author Comment.

Please also note the supplement to this comment:
https://www.atmos-chem-phys-discuss.net/acp-2019-347/acp-2019-347-AC2-supplement.pdf

---

## Author Comment (AC3) · 19 Aug 2019

**Reply to Comments from Reviewer #2**

We thank the reviewers for their valuable comments which help us improve the quality of the manuscript. We have carefully revised our manuscript following the reviewers' comments. Point-by-point responses are given below. The reviewers' comments are in black and our responses are in blue.

*Comment:*

A method section is missing. The authors may want to provide a Methodology section to cover the following items, how the literature search/review was conducted, what is the scope of the literature search, what are the primary source of publications (e.g. peer reviewed journal articles, government reports), restrictions if any (e.g. by year of publication, or by language).

*Response:*

We thank the reviewer for the suggestion. However, a method section is not quite common for a review paper. Considering the manuscript is already very long, we have not added a method section. Instead, we have modified the last paragraph of the Introduction part to make it more clear what the purpose of this review work is. Please refer to Lines 75–88 in the revised manuscript:

"Significant efforts have been made in the past decade for quantifying atmospheric Hg deposition through both direct observations and model simulations, especially on dry deposition (Lyman et al., 2009; Zhang et al., 2009; Holmes et al., 2011; Lai et al., 2011; Castro et al., 2012; Gustin et al., 2012; Peterson et al., 2012; L. Zhang et al., 2012; Fang et al., 2013; Sather et al., 2013; Lynam et al., 2014; Sather et al., 2014; Huang and Gustin, 2015a; Weiss-Penzias et al., 2016a; Zhang et al., 2016b; Hall et al., 2017; Sprovieri et al., 2017). Yet large uncertainties still exist due to limitations of current methods for Hg deposition measurements and modeling (Gustin et al., 2015). The purpose of this paper is to give an overview of the uncertainties in the observation and simulation of global speciated atmospheric Hg deposition to terrestrial surfaces. In this paper, we investigated results from the observation and simulation of global Hg

deposition, reviewed methods adopted for Hg deposition measurements and modeling, estimated the uncertainties of different methods for different Hg deposition forms, and summarized the overall uncertainty level of global Hg deposition."

*Comment:*

The scope of the review needs more justification. The title reads, "Global deposition of speciated atmospheric mercury to terrestrial surfaces: an overview". The rational of excluding the water surfaces (Figures 1, 6, 7 do include water through) and snow/ice over land should be presented.

*Response:*

We have modified the title and put focus on the uncertainties in the observation and simulation of global speciated atmospheric Hg deposition to terrestrial surfaces. Surface type is not our primary concern in the revised manuscript. The "water" surfaces here refer to the terrestrial surfaces near water. We have added the explanation to both figure captions and the main text. Please refer to Lines 146–147 in the revised manuscript:

"The "water" surfaces here refer to the terrestrial surfaces near water, e.g., coastal, offshore, and lakeside sites."

*Comment:*

The scientific contribution could be enhanced significantly. The manuscript as written is a somewhat descriptive presentation of estimation methods (sections 2 and 3) and Hg deposition values (sections 4 and 5). Consequently, there is a lack of new insights and findings. The authors are encouraged to conduct a rigorous research leading to more depth discussion that highlights the advancement, challenges, and directions for future research. Some potential topics are listed below (also see sample papers and a sample weblink at the end) 1) Comparison of co-located measurements with different techniques 2) Comparison of Hg deposition estimates by different models 3) Model-measurement comparison 4) Observed/predicted changes in Hg deposition due to changes in quantity of Hg emissions in local, regional or globe scale 5)

Observed/predicted changes in Hg deposition due to changes in profiles (e.g. the percentage of each Hg species in total emission) of Hg emissions in local, regional or globe scale 6) Contributions to observed/simulated Hg dry deposition from different sources or regions 7) The major sources of uncertainty in Hg deposition estimates and how to reduce those uncertainties 8) What is the knowledge or data gap (relevant to Hg deposition) that hinders our understand of the global Hg cycle, or the development and evaluation of emission control measures?

*Response:*

We greatly appreciate the valuable comment. We agree with the reviewer that the contribution of the manuscript was not clear. We have reorganized our manuscript, made significant revision, and added more discussion on the uncertainties in the observation and simulation of global speciated atmospheric Hg deposition to terrestrial surfaces. We believe the revised manuscript is more focused and more informative. Please refer to the revised manuscript.

*Comment:*

The "Bidirectional air-surface exchange model for GEM" is presented. However, dry deposition of GEM is estimated in many field studies and model simulations, including most GEM dry deposition data presented in the manuscript. Thus, the authors may want to include dry deposition models of GEM.

*Response:*

As mentioned in responses to previous comments, we have revised the manuscript extensively to focus on the uncertainties. The bidirectional model is a more commonly used model in recent years, so we have estimated the uncertainty in the simulation of GEM deposition flux based on the bidirectional model instead of the resistance model. Previous review work (Zhang et al., 2009) has discussed the two types of models in detail.

*Reference:*

*Zhang, L. M., Wright, L. P., and Blanchard, P.: A review of current knowledge concerning dry deposition of atmospheric mercury, Atmos. Environ., 43, 5853–5864,*

*10.1016/j.atmosenv.2009.08.019, 2009.*

*Comment:*

Please provide facts to support your statements, e.g. "For PBM dry deposition, a size-segregated resistance model is more and more widely applied" (L312)

*Response:*

The description here was not accurate. We have modified the expression. Please refer to Lines 700–703 in the revised manuscript:

"For PBM dry deposition, resistance models regarding both fine and coarse particles are more and more widely applied based on the theory that $v_d$ for atmospheric particles strongly depend on particle size (Dastoor and Larocque, 2004; Zhang et al., 2009; Zhang and He, 2014)."

*Comment:*

Most materials presented in sections 2 and 3 can be found in previous review/research papers, because those techniques have been around for a while. The authors could provide a summary table and direct the interested readers to those review/research papers, instead of a lengthy description of each method. Another option is to provide a comparative review of those methods and to include strength, weakness, recent advancements if any, and application issues.

*Response:*

We have modified the discussion on the methods for observation and simulation of Hg deposition to follow the estimation of uncertainties in the revised manuscript. Method details have been lessened.

*Comment:*

Section 4.3 (Forest deposition or Deposition over forests) could be better placed in section 5 (Global Hg deposition on different terrestrial surfaces).

*Response:*

We have reorganized the whole manuscript. The uncertainty analysis for forest Hg

deposition is based on methods for litterfall and throughfall deposition. Therefore, it is in parallel with wet and dry deposition. Please refer to Section 3.3 and 4.3 in the revised manuscript.

*Comment:*

If the authors decided to keep the equations, please 1) provide unit of each variable, 2) provide the source of each equation, 3) clarify the expansion factor in equations (8) and (9). Is it an expansion from a measurement in a small area to a forest? 4) explain how to calculate two resistances with equation (16).

*Response:*

We think variable unit is not quite necessary for the uncertainty analysis in this review work. We have added the sources of each equation and clarified the "expansion factor". Equation (16) means that there are two sets of $v_g$ (gravitational settling velocity), $R_a$ (aerodynamic resistance), and $R_s$ (surface resistance) for fine and coarse particles, respectively.

*Comment:*

Please state the mechanism of Hg deposition via cloud/fog at high elevation sites (L258).

*Response:*

Cloud and fog can scavenge Hg in the atmosphere. At alpine or coastal sites, Hg can deposit onto the ground through cloud or fog. Cloud or fog is not able to be collected by precipitation samplers. Studies (Stankwitz et al., 2012; Weiss-Penzias et al., 2016b; Gerson et al., 2017) have shown that cloud and fog water have higher Hg concentration than rain water in the same region, and cloud and fog could have a remarkable contribution to Hg wet deposition in high-elevation forests and near-water surfaces.

*References:*

*Stankwitz, C., Kaste, J. M., and Friedland, A. J.: Threshold increases in soil lead and mercury from tropospheric deposition across an elevational gradient, Environ. Sci. Technol., 46, 8061–8068, 10.1021/es204208w, 2012.*

*Weiss-Penzias P., Coale K, Heim W, Fernandez D, Oliphant A, Dodge C, Hoskins D,*

*Farlin J, Moranville R, Olson A. Total- and monomethyl-mercury and major ions in coastal California fog water: Results from two years of sampling on land and at sea. Elem. Sci. Anth., 4, 1–18, 10.12952/journal.elementa.000101, 2016b.*

*Gerson, J. R., Driscoll, C. T., Demers, J. D., Sauer, A. K., Blackwell, B. D., Montesdeoca, M. R., Shanley, J. B., and Ross, D. S.: Deposition of mercury in forests across a montane elevation gradient: Elevational and seasonal patterns in methylmercury inputs and production, J. Geophys. Res. Biogeo., 122, 1922–1939, 10.1002/2016jg003721, 2017.*

*Comment:*

L456, the authors may want to distinguish the net emission fluxes from "natural GEM emission sources".

*Response:*

We have modified the expression according to the comment. Please refer to Lines 217–220 in the revised manuscript:

"The four Asian sites using micrometeorological methods all show negative values $(-36.3\pm19.6\,\mu g\,m^{-2}\,yr^{-1})$, indicating the role of East Asia as a net emission source rather than a net deposition sink (Luo et al., 2014; Luo et al., 2016; Ci et al., 2016; Yu et al., 2018)."

*Comment:*

Figure 6, "precipitation levels" or "annual precipitation"?

*Response:*

We have modified to wording accordingly.

*Comment:*

The papers from which data were obtained to generate each figure could be tabulated and presented as Supplement Information.

*Response:*

We have tabulated the raw data and created a Supporting Information file.

*Comment:*

There are quite a few awkward sentences and word choices, e.g. "*Ci* is the total Hg concentration in precipitation water" (L193), "Usually, GOM and PBM contribute equivalently to Hg wet deposition (Cheng et al., 2015)." (L206), add "GEM dry deposition is equivalent to GOM and PBM dry deposition, even significantly higher than in forests" (L535), "consequently exhibit significantly high litterfall Hg deposition fluxes." (L560), "Water surfaces could affect Hg wet deposition through fog scavenging." (L580), "The contribution GEM dry deposition has been underestimated previously." (L596), "Cloud, fog or even dew Hg deposition needs careful investigation" (L599), please rephrase.

*Response:*

We thank the reviewer for the detailed comments. We have rephrased or deleted these sentences. Please refer to the revised manuscript.

*Comment:*

There are some contradicting or confusing statements, e.g. "Based on available measurements of PBM size distributions and fine/coarse PBM mass ratios, Zhang et al. (2016b) assumed 30% of the total PBM mass to be coarse particles in order to estimate total PBM dry deposition flux based on the theory that PBM has the same proportion in both fine and course particles." (L318)

*Response:*

We have modified the statement. Please refer to Lines 706–709 in the revised manuscript:

"Based on measurements of particle size distributions and Hg mass distribution between fine and coarse particles, Zhang et al. (2016b) assumed that coarse particles account for 30 % of the total PM, and the Hg mass concentrations on fine and coarse particles are consistent."

*Comment:*

Avoid the use of first person, i.e. "we".

*Response:*

We have avoided the use of "we".

---

## Author Comment (AC4) · 19 Aug 2019

**Reply to Comments from Reviewer #3**

We thank the reviewers for their valuable comments which help us improve the quality of the manuscript. We have carefully revised our manuscript following the reviewers' comments. Point-by-point responses are given below. The reviewers' comments are in black and our responses are in blue.

*Comment:*

In general, this paper is not easy to follow, the authors jump from one topic to another. They did not do advanced discussion. In more paragraphs, they only described methods and data, and probably two/three sentences to summarize/discuss what they learn from these methods/data. There is nothing inspiring readers. A review paper should do better than that.

Their conclusions/summaries are not new. Gustin's group has published couple review articles discussing the first three aspects in 2015, and the 4th aspect has been mentioned in multiple previous articles. I really do not find any new concepts in this article, and how can we solve the difficulties that the Hg research community is facing. For example, do the authors have any suggestion to understand behaviors of various GOM compounds in the atmosphere?

I agree this is an important research field and there are gaps which make scientists cannot fully understand global Hg cycle. A review paper related to this topic should be published to draw attention from environmental research groups. However, the way that this paper is done cannot provide useful information to scientists. I suggest the authors re-think about the article structure and put more efforts on advanced discussions.

*Response:*

We greatly appreciate the valuable comment. We agree with the reviewer that the contribution of the manuscript was not clear. We have reorganized our manuscript, made significant revision, and added more discussion on the uncertainties in the observation and simulation of global speciated atmospheric Hg deposition to terrestrial surfaces. We believe the revised manuscript is more focused and more informative.

Please refer to the revised manuscript.

*Comment:*

Abstract is read more like a summary than an abstract. I suggest to re-write the abstract and focus on your key aspects. Moreover, the authors must provide some potential solutions/suggests for each gap that are discussed in their conclusions.

*Response:*

We have rewritten the abstract based on the revised manuscript. Here is our updated abstract:

"One of the most important processes in the global mercury (Hg) biogeochemical cycling is the deposition of atmospheric Hg, including gaseous elemental mercury (GEM), gaseous oxidized mercury (GOM), and particulate-bound mercury (PBM), to terrestrial surfaces. Results of wet, dry, and forest Hg deposition from global observation networks, individual monitoring studies, and observation-based simulations have been reviewed in this study. Uncertainties in the observation and simulation of global speciated atmospheric Hg deposition to terrestrial surfaces have been systemically estimated based on assessment of commonly used observation methods, campaign results for comparison of different methods, model evaluation with observation data, and sensitivity analysis for model parameterization. The uncertainties of GOM and PBM dry deposition measurements come from the interference of unwanted Hg forms or incomplete capture of targeted Hg forms, while that of GEM dry deposition observation originates from the lack of standardized experimental system and operating procedure. The large biases in the measurements of GOM and PBM concentration and the high sensitivities of key parameters in resistance models lead to high uncertainties in GOM and PBM dry deposition simulation. Non-precipitation Hg wet deposition could play a crucial role in alpine and coastal regions, and its high uncertainties in both observation and simulation affect the overall uncertainties of Hg wet deposition. The overall uncertainties in the observation and simulation of the total global Hg deposition were estimated to be $\pm(30–50)$ % and $\pm(50–70)$ %, respectively, with the largest contributions from dry deposition. According to the results from

uncertainty analysis, future research needs were recommended, among which global Hg dry deposition network, unified methods for GOM and PBM dry deposition measurements, quantitative methods for GOM speciation, campaigns for comprehensive forest Hg behavior, and more efforts on long-term Hg deposition monitoring in Asia are the top priorities."

*Comment:*

Introduction is fine, but this is a review paper. There are more previous Hg review articles, such as Selin et al., 2007, and some key finding paper are not included in this review paper, such as Moore et al., 2014 Nature. These articles might not be directly linked to Hg deposition, but they do have indirect impacts on Hg deposition. After reading this article, I feel the authors focus on the measuring methods and numeric models, but do not discuss in advance about global deposition processes.

*Response:*

We have sharpened the discussion in the manuscript to focus on the uncertainties in the observation and simulation of global speciated atmospheric Hg deposition to terrestrial surfaces. We have also added the recent modeling work for Hg wet deposition. Please refer to Section 4.1.1 in the revised manuscript.

*Comment:*

A summary table or multiple summary tables would help the readers to read through this section.

*Response:*

We have added a summary table for the uncertainties discussed in this study. Please refer to Table 1. We have also created a Supporting Information file listing all the Hg deposition studies.

*Comment:*

Surrogate surface: the key point of this method is the surface affinity and fluent conditions near surface, but I did not see the authors discuss these here. Huang et al.,

2011 published a paper discussing fluent conditions near KSS surface, and how this impacts mass transfer.

*Response:*

We have added more discussion on how the sampler designs or fluent conditions affect the uncertainty of the surrogate surface method. Please refer to Lines 382–414 in the revised manuscript:

"Different surrogate surfaces were used to measure different RM forms. Mounts with cation-exchange membranes (CEMs) are widely used for GOM dry deposition measurements (Lyman et al., 2007; Lyman et al., 2009; Castro et al., 2012; Huang et al., 2012a; Peterson et al., 2012; Sather et al., 2013). The down-facing aerodynamic mount with CEM is considered to be the most reliable deployment for GOM dry deposition measurements so far (Lyman et al., 2009; Huang et al., 2014). Knife-edge surrogate surface (KSS) samplers with quartz fiber filter (QFFs) and dry deposition plates (DDPs) were deployed for PBM dry deposition measurements (Lai et al., 2011; Fang et al., 2012b; Fang et al., 2013). However, these samplers are not well verified to reflect the deposition velocity of PBM, and hence not widely accepted. KCl-coated QFFs were used to measure the total RM (GOM+PBM) dry deposition, but failed to capture GOM efficiently (Lyman et al., 2009; Lai et al., 2011).

According to Eq. (4), the uncertainty of RM dry deposition comes from the uncertainties of RM concentration and dry deposition velocity. The uncertainty of RM concentration mainly originates from the interference of unwanted RM forms or incomplete capture of targeted RM forms. CEMs exhibited a GOM capture rate of 51–107 % in an active sampling system (Huang and Gustin, 2015b). The CEM mounts designed to measure only GOM dry deposition capture part of fine PBM (Lyman et al., 2009; Huang et al., 2014), while the KSS samplers with QFFs designed to measure only PBM dry deposition may also collect part of GOM (Rutter and Schauer, 2007; Gustin et al., 2015). Based on the RM concentration measurements and the surrogate surface method evaluations, the GOM concentration related uncertainty is estimated to be ±50 % (Lyman et al., 2009; Lyman et al., 2010; Gustin et al., 2012; Fang et al., 2013; Zhang et al., 2013; Huang et al., 2014). The design of the sampler (e.g., the sampler orientation,

the shape of the sampler, variation in turbulence, low surface resistances, passivation, etc.) leads to the dry deposition velocity related uncertainty which is about ±50 % for GOM (Lyman et al., 2009; Lai et al., 2011; Huang et al., 2012a). Calculating based on the method described by Eq. (2), the overall uncertainty of GOM dry deposition observation is ±70 %. There is not enough information to quantify the overall uncertainty of PBM dry deposition observation in a similar way. Based on the distribution of daily samples in the study of Fang et al. (2012b), the overall uncertainty of PBM dry deposition measurements is assumed to be roughly ±100 % or within a factor of 2."

*Comment:*

Enclosure methods: Choi and Holsen 2008/2009 articles are also important, and the authors did not discuss about the bio-process/photo-process related to Hg reduction in DFC.

*Response:*

We have added the discussion of the influence of DFC material based on the study of Choi and Holsen (2009). Please refer to Lines 474–476 in the revised manuscript:

"Choi and Holsen (2009) reported that the polycarbonate DFC blocks most of the UV-B light from reaching the soil where $Hg^{2+}$ can be reduced to $Hg^0$, and hence the GEM emission flux might be underestimated by at most 20 %."

*Comment:*

Micrometeorological methods: This method has been used to understand GOM flux as well, but no discussion here.

*Response:*

We have added discussion on micrometeorological methods for GOM dry deposition measurement. Please refer to Lines 372–375 in the revised manuscript:

"The micrometeorological methods and the enclosure methods were also adopted in some studies (Poissant et al., 2004; Zhang et al., 2005; Skov et al., 2006), but not widely used due to the high uncertainties in the measurements of GOM and PBM

concentrations using the Tekran system."

*Comment:*

In forests: Choi and Holsen 2009, and there are more articles from Driscoll's group discussing Hg cycle in forests.

*Response:*

We have cited the study of Choi and Holsen (2009). We have also cited articles from Driscoll's group:

*Blackwell, B. D., and Driscoll, C. T.: Using foliar and forest floor mercury concentrations to assess spatial patterns of mercury deposition, Environ. Pollut., 202, 126–134, 10.1016/j.envpol.2015.02.036, 2015a.*

*Blackwell, B. D., and Driscoll, C. T.: Deposition of mercury in forests along a montane elevation gradient, Environ. Sci. Technol., 49, 5363–5370, 10.1021/es505928w, 2015b.*

*Bushey, J. T., Nallana, A. G., Montesdeoca, M. R., and Driscoll, C. T.: Mercury dynamics of a northern hardwood canopy, Atmos. Environ., 42, 6905–6914, 10.1016/j.atmosenv.2008.05.043, 2008.*

*Gerson, J. R., Driscoll, C. T., Demers, J. D., Sauer, A. K., Blackwell, B. D., Montesdeoca, M. R., Shanley, J. B., and Ross, D. S.: Deposition of mercury in forests across a montane elevation gradient: Elevational and seasonal patterns in methylmercury inputs and production, J. Geophys. Res. Biogeo., 122, 1922–1939, 10.1002/2016jg003721, 2017.*

*Luo, Y., Duan, L., Driscoll, C. T., Xu, G. Y., Shao, M. S., Taylor, M., Wang, S. X., and Hao, J. M.: Foliage/atmosphere exchange of mercury in a subtropical coniferous forest in south China, J. Geophys. Res. Biogeo., 121, 2006–2016, 10.1002/2016jg003388, 2016.*

*Comment:*

GOM resistance: page 10 line 299-310, Gustin et al., 2015 has summarized this, this is not a new idea. I just feel, the authors are writing a review article, but they are repeating the concepts from the summaries in other's review articles without adding their new

thoughts.

*Response:*

We have sharpened the discussion in the manuscript to focus on the uncertainties in the observation and simulation of global speciated atmospheric Hg deposition to terrestrial surfaces.

*Comment:*

Page 13 line 401-402, is ambient concentrations not important?

*Response:*

We have deleted this sentence.

*Comment:*

Page 14, line 412-414, Europe has…., any ambient data to support this argument?

*Response:*

We have deleted this argument.

*Comment:*

Line 427, deposition fluxes concentrations, what does "fluxes concentrations" mean?

*Response:*

We have modified this statement. Please refer to Lines 182–183 in the revised manuscript:

"Most studies on GOM dry deposition were conducted in North America and Europe"

*Comment:*

Line 435-439, the authors should explain why they are showing significantly different? Different surface affinity?

*Response:*

We have discussed the surrogate surface method in detail. Please refer to Lines 382–414 in the revised manuscript:

"Different surrogate surfaces were used to measure different RM forms. Mounts with

cation-exchange membranes (CEMs) are widely used for GOM dry deposition measurements (Lyman et al., 2007; Lyman et al., 2009; Castro et al., 2012; Huang et al., 2012a; Peterson et al., 2012; Sather et al., 2013). The down-facing aerodynamic mount with CEM is considered to be the most reliable deployment for GOM dry deposition measurements so far (Lyman et al., 2009; Huang et al., 2014). Knife-edge surrogate surface (KSS) samplers with quartz fiber filter (QFFs) and dry deposition plates (DDPs) were deployed for PBM dry deposition measurements (Lai et al., 2011; Fang et al., 2012b; Fang et al., 2013). However, these samplers are not well verified to reflect the deposition velocity of PBM, and hence not widely accepted. KCl-coated QFFs were used to measure the total RM (GOM+PBM) dry deposition, but failed to capture GOM efficiently (Lyman et al., 2009; Lai et al., 2011).

According to Eq. (4), the uncertainty of RM dry deposition comes from the uncertainties of RM concentration and dry deposition velocity. The uncertainty of RM concentration mainly originates from the interference of unwanted RM forms or incomplete capture of targeted RM forms. CEMs exhibited a GOM capture rate of 51–107 % in an active sampling system (Huang and Gustin, 2015b). The CEM mounts designed to measure only GOM dry deposition capture part of fine PBM (Lyman et al., 2009; Huang et al., 2014), while the KSS samplers with QFFs designed to measure only PBM dry deposition may also collect part of GOM (Rutter and Schauer, 2007; Gustin et al., 2015). Based on the RM concentration measurements and the surrogate surface method evaluations, the GOM concentration related uncertainty is estimated to be ±50 % (Lyman et al., 2009; Lyman et al., 2010; Gustin et al., 2012; Fang et al., 2013; Zhang et al., 2013; Huang et al., 2014). The design of the sampler (e.g., the sampler orientation, the shape of the sampler, variation in turbulence, low surface resistances, passivation, etc.) leads to the dry deposition velocity related uncertainty which is about ±50 % for GOM (Lyman et al., 2009; Lai et al., 2011; Huang et al., 2012a). Calculating based on the method described by Eq. (2), the overall uncertainty of GOM dry deposition observation is ±70 %. There is not enough information to quantify the overall uncertainty of PBM dry deposition observation in a similar way. Based on the distribution of daily samples in the study of Fang et al. (2012b), the overall uncertainty

of PBM dry deposition measurements is assumed to be roughly ±100 % or within a factor of 2."

*Comment:*

Page 17, line 537-540, different surface (e.g. forest vs grassland), there are many differences between these two surface types, such as leaf area index, but the authors just simply summarized all these difference depositions based on chemistry and not talking about the characteristic of surfaces.

*Response:*

High GOM concentration at high elevation leads to high GOM deposition. Leaf area index (LOI) also has impact on GOM dry deposition, but not as much. We have added the uncertainty analysis in the simulation of GOM dry deposition with resistance model. Please refer to Section 4.2.2 in the revised manuscript.

---

## Author Comment (AC5) · 19 Aug 2019

**Supporting Information for the manuscript entitled:**

**Uncertainties in the observation and simulation of global speciated atmospheric mercury deposition to terrestrial surfaces**

**Lei Zhang[1,2,*], Peisheng Zhou[1], Shuzhen Cao[1], and Yu Zhao[1,2]**

[1] School of the Environment, Nanjing University, 163 Xianlin Avenue, Nanjing, Jiangsu 210023, China

[2] State Key Laboratory of Pollution Control and Resource Reuse, Nanjing University, 163 Xianlin Avenue, Nanjing, Jiangsu 210023, China

*Correspondence to:* Lei Zhang (lzhang12@nju.edu.cn)

Table S1: Hg wet deposition distribution along different land use type (B: Barren; C/N: Cropland/Natural Vegetation mosaics; C: Croplands; DB: Deciduous Broadleaf Forests; DN: Deciduous Needleleaf Forests; EB: Evergreen Broadleaf Forests; EN: Evergreen Needleleaf Forests; MIX: Mixed Forests; G: Grasslands; PW: Permenant Wetlands; S: Savannas; WS: Woody Savannas; OS: Open Shrublands; Urban and Built-up Lands; W: Water.)

| Site Name | Network | LUC | Latitude | Longitude | Period | Conc. (ng/L) | Prec. (mm) | Flux (µg m$^{-2}$ yr$^{-1}$) | References |
|---|---|---|---|---|---|---|---|---|---|
| Craters of the MNM | MDN | B | 43.46 | -113.56 | 2008-2015 | 12.39 | 434.2 | 5.1 | |
| Patricia McInnes | | B | 56.75 | -111.48 | 2011 | 11.2 | 98 | 1.1 | Lynam et al., 2017 |
| Ny-Ålesund | GMOS | B | 78.90 | 11.88 | 2012–2015 | 5.9 | 232.3 | 1.1 | Sprovieri et al., 2017 |
| Dixon Springs A.C. | MDN | C/N | 37.44 | -88.67 | 2008-2015 | 9.84 | 1285.2 | 12.3 | |
| Ann Arbor | MDN | C/N | 42.42 | -83.90 | 2008-2015 | 12.4 | 864.9 | 10.6 | |
| Millersville | MDN | C/N | 39.99 | -76.39 | 2008-2015 | 7.4 | 1131.3 | 8.3 | |
| Longview | MDN | C/N | 32.38 | -94.71 | 2008-2015 | 9.6 | 1189.1 | 11.2 | |
| Harcum | MDN | C/N | 37.53 | -76.49 | 2008-2015 | 7.1 | 1243.2 | 8.8 | |
| Lake Geneva | MDN | C/N | 42.58 | -88.50 | 2008-2015 | 10.8 | 965.2 | 10.3 | |
| Puding | | C/N | 26.37 | 105.80 | 2005/05-2006/07 | 20.6 | 1203 | 24.8 | Guo et al., 2008 |
| Hongjiadu | | C/N | 26.88 | 105.85 | 2005/05-2006/08 | 39.4 | 881 | 34.7 | Guo et al., 2008 |
| Dongfeng | | C/N | 26.85 | 106.13 | 2005/05-2006/10 | 37.4 | 970 | 36.3 | Guo et al., 2008 |
| Waldhof | EMEP | C/N | 52.80 | 10.75 | 2009 | | 638 | 2.9 | Bieser et al., 2014 |
| Yarner Wood | EMEP | C/N | 50.59 | -3.70 | 2009 | | 1296 | 3.3 | Bieser et al., 2014 |
| Genesee | MDN | C | 53.30 | -114.20 | 2008-2015 | 11.2 | 390.4 | 4.2 | |
| Deer Flats | MDN | C | 43.55 | -116.64 | 2008-2015 | 10.56 | 215.3 | 2.3 | |
| Bondville | MDN | C | 40.05 | -88.37 | 2008-2015 | 11.43 | 917.8 | 10.5 | |
| Clifty Falls State Park | MDN | C | 38.76 | -85.42 | 2008-2015 | 10.23 | 1387 | 14.1 | |
| Southwest Purdue A.C. | MDN | C | 38.74 | -87.49 | 2008-2015 | 8.8 | 1216.6 | 10.7 | |
| Reserve | MDN | C | 39.98 | -95.57 | 2008-2015 | 14.4 | 825 | 11.8 | |
| West Mineral | MDN | C | 37.27 | -94.94 | 2008-2015 | 12.2 | 1097.9 | 13.2 | |
| Coffey County Lake | MDN | C | 38.20 | -95.66 | 2008-2015 | 12.7 | 955.9 | 11.9 | |
| Glen Elder State Park | MDN | C | 39.51 | -98.34 | 2008-2015 | 16.6 | 610.5 | 10.0 | |

| Site Name | Network | LUC | Latitude | Longitude | Period | Conc. (ng/L) | Prec. (mm) | Flux (µg m$^{-2}$ yr$^{-1}$) | References |
|---|---|---|---|---|---|---|---|---|---|
| Chase | MDN | C | 32.10 | -91.71 | 2008-2015 | 7.2 | 1726.1 | 12.4 | |
| Alexandria | MDN | C | 31.17 | -92.40 | 2008-2015 | 9.4 | 1225.5 | 11.4 | |
| Caribou | MDN | C | 46.87 | -68.01 | 2008-2015 | 5.8 | 1092.6 | 7.0 | |
| Kellogg | MDN | C | 42.41 | -85.39 | 2008-2015 | 8.5 | 1048 | 8.9 | |
| Lamberton | MDN | C | 44.24 | -95.30 | 2008-2015 | 12.6 | 661.4 | 8.3 | |
| Lostwood | MDN | C | 48.64 | -102.40 | 2008-2015 | 11.4 | 333.5 | 3.8 | |
| Mead | MDN | C | 41.15 | -96.49 | 2008-2015 | 15.2 | 809.3 | 12.0 | |
| Miami | MDN | C | 36.90 | -94.76 | 2008-2015 | 11.9 | 1189.2 | 14.2 | |
| Egbert | MDN | C | 44.23 | -79.79 | 2008-2015 | 8.0 | 755.7 | 6.2 | |
| Arendtsville | MDN | C | 39.92 | -77.31 | 2008-2015 | 7.2 | 1144.8 | 8.3 | |
| Valley Forge | MDN | C | 40.12 | -75.88 | 2008-2015 | 7.0 | 1214.1 | 8.4 | |
| Eagle Butte | MDN | C | 44.99 | -101.24 | 2008-2015 | 14.1 | 498.1 | 6.8 | |
| Bratt's Lake BSRN | MDN | C | 50.20 | -104.71 | 2008-2015 | 8.2 | 405 | 3.3 | |
| Horicon Marsh | MDN | C | 43.47 | -88.62 | 2008-2015 | 9.8 | 917.5 | 9.0 | |
| Three Gorges | | C | 29.90 | 107.50 | 2012/12–2013/10 | 18 | 722 | 13.0 | Zhao et al., 2015 |
| Yucheng | | C | 36.95 | 116.60 | 2012-2013 | | 528 | 8.8 | Sommar et at., 2016 |
| Seine Bay | | C | 49.40 | 0.50 | 2010/10-1012/03 | | 802.3 | 9.0 | Connan et al., 2013 |
| Zingst | EMEP | C | 54.43 | 12.73 | 2009 | | 561 | 3.4 | Bieser et al., 2014 |
| Heigham Holmes | EMEP | C | 52.72 | -1.62 | 2009 | | 462 | 2.2 | Bieser et al., 2014 |
| Auchencorth Moss | EMEP | C | 55.78 | 3.23 | 2009 | | 679 | 3.4 | Bieser et al., 2014 |
| Banchory | EMEP | C | 57.07 | -2.53 | 2009 | | 777 | 3.8 | Bieser et al., 2014 |
| Kollumerwaard | EMEP | C | 52.30 | 4.50 | 2009 | | 719 | 2.8 | Bieser et al., 2014 |

| Site Name | Network | LUC | Latitude | Longitude | Period | Conc. (ng/L) | Prec. (mm) | Flux (µg m$^{-2}$ yr$^{-1}$) | References |
|---|---|---|---|---|---|---|---|---|---|
| Mammoth Cave | MDN | DB | 37.13 | -86.15 | 2008-2015 | 8.1 | 1362.4 | 9.5 | |
| Piney Reservoir | MDN | DB | 39.71 | -79.01 | 2008-2015 | 7.0 | 1036.7 | 7.3 | |
| Carrabassett Valley | MDN | DB | 45.08 | -70.21 | 2008-2015 | 6.7 | 1291.9 | 8.6 | |
| Greenville Station | MDN | DB | 45.49 | -69.66 | 2008-2015 | 5.3 | 1301 | 6.9 | |
| Douglas Lake | MDN | DB | 45.56 | -84.68 | 2008-2015 | 6.9 | 943.23 | 6.6 | |
| Marcell | MDN | DB | 47.53 | -93.47 | 2008-2015 | 9.1 | 751.3 | 17.7 | |
| Ashland | MDN | DB | 38.75 | -92.20 | 2008-2015 | 10.9 | 1044.5 | 11.2 | |
| Mingo | MDN | DB | 36.97 | -90.14 | 2008-2015 | 10.0 | 1443.2 | 14.2 | |
| Huntington Wildlife | MDN | DB | 43.97 | -74.22 | 2008-2015 | 5.1 | 1136.8 | 5.8 | |
| Biscuit Brook | MDN | DB | 41.99 | -74.50 | 2008-2015 | 5.5 | 1565.3 | 8.7 | |
| Allegheny | MDN | DB | 40.46 | -78.56 | 2008-2015 | 8.1 | 1096.3 | 8.8 | |
| Young Woman's Creek | MDN | DB | 41.41 | -77.68 | 2008-2015 | 7.9 | 1020.8 | 8.0 | |
| Goddard State Park | MDN | DB | 41.43 | -80.15 | 2008-2015 | 8.9 | 1095.5 | 9.7 | |
| Kane Experimental Forest | MDN | DB | 41.60 | -78.77 | 2008-2015 | 7.2 | 1205.9 | 8.6 | |
| Leading Ridge | MDN | DB | 40.66 | -77.94 | 2008-2015 | 7.7 | 1015.9 | 7.8 | |
| Little Pine State Park | MDN | DB | 41.36 | -77.36 | 2008-2015 | 7.6 | 970.6 | 7.4 | |
| Milford | MDN | DB | 41.33 | -74.82 | 2008-2015 | 6.9 | 1255.6 | 8.7 | |
| Congaree Swamp | MDN | DB | 33.81 | -80.78 | 2008-2015 | 9.0 | 1217.5 | 10.9 | |
| Great Smoky Mountains | MDN | DB | 35.66 | -83.59 | 2008-2015 | 9.6 | 1547.4 | 13.3 | |
| Shenandoah | MDN | DB | 38.52 | -78.43 | 2008-2015 | 5.9 | 1493.4 | 8.6 | |
| Underhill | MDN | DB | 44.53 | -72.87 | 2008-2015 | 5.3 | 1354.6 | 7.2 | |
| Brule River | MDN | DB | 46.75 | -91.61 | 2008-2015 | 9.0 | 812.9 | 7.4 | |
| Potawatomi | MDN | DB | 45.56 | -88.81 | 2008-2015 | 8.3 | 865.5 | 7.3 | |

| Site Name | Network | LUC | Latitude | Longitude | Period | Conc. (ng/L) | Prec. (mm) | Flux (µg m$^{-2}$ yr$^{-1}$) | References |
|---|---|---|---|---|---|---|---|---|---|
| Devil's Lake | MDN | DB | 43.44 | -89.68 | 2008-2015 | 10.1 | 1058.5 | 10.5 | |
| Trout Lake | MDN | DB | 46.05 | -89.65 | 2008-2015 | 9.4 | 829.1 | 7.7 | |
| Canaan Valley Institute | MDN | DB | 39.12 | -79.45 | 2008-2015 | 7.0 | 1307.3 | 9.1 | |
| Mt. Damei | | DB | 29.63 | 121.57 | 2012/08-2014/08 | 3.7 | 1622 | 6.0 | Fu et al., 2016a |
| Han River | | DB | 37.53 | 127.30 | 2008/02-2010/02 | 12.6 | 1085.9 | 4.3 | Han et al., 2016 |
| Tieshanping | | DN | 29.62 | 106.69 | 2005/03-2006/03 | 32.3 | 898 | 29.0 | Wang et al., 2009 |
| Oak Grove | MDN | EB | 30.98 | -88.93 | 2008-2015 | 9.9 | 1705 | 16.0 | |
| Mt. Ailao | GMOS | EB | 24.54 | 101.03 | 2011–2014 | 4.5 | 896.7 | 3.3 | Sprovieri et al., 2017 |
| Xiaoping,Xiamen | | EB | 24.85 | 118.04 | 2012/06-2013/05 | 11.3 | 1271 | 14.4 | Xu et al., 2014 |
| Bantou,Xiamen | | EB | 24.67 | 118.03 | 2012/06-2013/05 | 14 | 1252 | 17.6 | Xu et al., 2014 |
| Chongqing | | EB | 29.83 | 106.38 | 2008-2015 | 34.25 | 1104 | 37.8 | Qin et al., 2016 |
| Saturna Island | MDN | EN | 48.78 | -123.13 | 2008-2015 | 6.2 | 820.31 | 5.1 | |
| Sequoia | MDN | EN | 36.57 | -118.78 | 2008-2015 | 7.173 | 842.3 | 5.5 | |
| H. J. Andrews | MDN | EN | 44.21 | -122.26 | 2008-2015 | 3.8 | 2436.6 | 9.2 | |
| Chapais | MDN | EN | 49.82 | -74.98 | 2008-2015 | 5.3 | 1064.1 | 5.6 | |
| Makah | MDN | EN | 48.29 | -124.65 | 2008-2015 | 3.6 | 1888.7 | 6.9 | |
| Mt. Gongga | | EN | 29.65 | 102.12 | 2006 | 9.9 | 919 | 9.1 | Fu et al., 2008 |
| Mt. Gongga | | EN | 29.58 | 101.93 | 2005/05-2006/4 | 14.3 | 1825 | 26.1 | Fu et al., 2010a |
| SET station | | EN | 29.77 | 94.73 | 2010/05-2012/10 | 4 | 975 | 3.9 | Huang et al., 2015 |
| Qianyanzhou | | EN | 26.75 | 115.07 | 2013/12-2014/11 | 12.5 | 1472 | 18.4 | Cheng et al., 2017 |
| Huitong | | EN | 26.83 | 109.75 | 2013/12-2014/11 | 22.2 | 703 | 15.6 | Cheng et al., 2017 |
| Schmücke | EMEP | EN | 50.65 | 10.77 | 2009 | | 1310 | 3.1 | Bieser et al., 2014 |
| Iskbra | GMOS | EN | 45.56 | 14.86 | 2011–2015 | 5.6 | 1203.4 | 6.7 | Sprovieri et al., 2017 |

| Site Name | Network | LUC | Latitude | Longitude | Period | Conc. (ng/L) | Prec. (mm) | Flux (µg m$^{-2}$ yr$^{-1}$) | References |
|---|---|---|---|---|---|---|---|---|---|
| Bariloche | GMOS | EN | -41.43 | -71.42 | 2014–2015 | 0.5 | 549.5 | 0.3 | Sprovieri et al., 2017 |
| North Atlantic Coastal Lab | MDN | MIX | 41.98 | -70.02 | 2008-2015 | 4.6 | 1204.6 | 5.5 | |
| Bridgton | MDN | MIX | 44.11 | -70.73 | 2008-2015 | 5.7 | 1257.3 | 7.2 | |
| Acadia | MDN | MIX | 44.38 | -68.26 | 2008-2015 | 4.8 | 1509.1 | 7.2 | |
| Fernberg | MDN | MIX | 47.95 | -91.50 | 2008-2015 | 8.3 | 726 | 5.7 | |
| Waccamaw | MDN | MIX | 34.26 | -78.48 | 2008-2015 | 7.4 | 1347.7 | 9.8 | |
| Candor | MDN | MIX | 35.26 | -79.84 | 2008-2015 | 6.2 | 1203.4 | 7.4 | |
| Pettigrew State Park | MDN | MIX | 35.74 | -76.51 | 2008-2015 | 7.2 | 1309.4 | 9.3 | |
| Kejimkujik National Park | MDN | MIX | 44.43 | -65.21 | 2008-2015 | 4.6 | 1569.1 | 7.1 | |
| Mt. Changbai | GMOS | MIX | 42.40 | 128.11 | 2011–2014 | 28.3 | 327.9 | 2.5 | Sprovieri et al., 2017 |
| Mt. Leigong | | MIX | 26.39 | 108.20 | 2008/05-2009/05 | 4 | 1525 | 6.1 | Fu et al., 2016a |
| Mt. Simian,Chongqing | | MIX | 28.62 | 106.40 | 2012/03-2013/02 | 10.94 | 1412.2 | 15.45 | Ma et al., 2016 |
| Deokjeok | | MIX | 37.23 | 126.12 | 2007-2008 | 12.4 | 476.3 | 6.9 | Nguyen et al., 2016 |
| Chuncheon | | MIX | 37.95 | 127.81 | 2007 | 11.425 | 1090 | 12.5 | Ahn et al., 2011 |
| Schauinsland | EMEP | MIX | 47.90 | 7.90 | 2009 | | 1382 | 5.1 | Bieser et al., 2014 |
| Birkenes | EMEP | MIX | 58.38 | 8.25 | 2009 | | 1790 | 3.8 | Bieser et al., 2014 |
| Vavihill | EMEP | MIX | 56.02 | 13.15 | 2009 | | 605 | 3.6 | Bieser et al., 2014 |
| Longobucco | GMOS | MIX | 39.39 | 16.61 | 2012–2013 | 5.3 | 282.1 | 1.7 | Sprovieri et al., 2017 |
| Henry Kroeger | MDN | G | 51.42 | -110.83 | 2008-2015 | 10.8 | 298.2 | 3.2 | |
| Kodiak | MDN | G | 57.72 | -152.56 | 2008-2015 | 2.1 | 2371.6 | 4.8 | |
| Sycamore Canyon | MDN | G | 35.14 | -111.97 | 2008-2015 | 24.5 | 353.99 | 7.7 | |
| Converse Flats | MDN | G | 34.19 | -116.91 | 2008-2015 | 9.2 | 492.3 | 4.4 | |
| Buffalo Pass - Summit Lake | MDN | G | 40.54 | -106.68 | 2008-2015 | 8.68 | 1050.4 | 9.2 | |

| Site Name | Network | LUC | Latitude | Longitude | Period | Conc. (ng/L) | Prec. (mm) | Flux (µg m$^{-2}$ yr$^{-1}$) | References |
|---|---|---|---|---|---|---|---|---|---|
| Mesa | MDN | G | 37.20 | -108.49 | 2008-2015 | 23.01 | 438 | 9.4 | |
| Everglades N.P.R.C | MDN | G | 25.39 | -80.68 | 2008-2015 | 13.35 | 1435.8 | 19.0 | |
| McCall | MDN | G | 44.89 | -116.10 | 2008-2015 | 8.09 | 651.8 | 5.3 | |
| Lake Scott State Park | MDN | G | 38.67 | -100.92 | 2008-2015 | 15.9 | 520.9 | 8.1 | |
| Cimarron | MDN | G | 37.13 | -101.82 | 2008-2015 | 17.1 | 354.6 | 6.1 | |
| Badger Peak | MDN | G | 45.63 | -106.55 | 2008-2015 | 13.2 | 400.3 | 5.3 | |
| Santee | MDN | G | 42.83 | -97.85 | 2008-2015 | 13.7 | 595.4 | 8.1 | |
| Lesperance Ranch | MDN | G | 41.50 | -117.50 | 2008-2015 | 34.0 | 165 | 2.7 | |
| Gibb's Ranch | MDN | G | 41.57 | -115.21 | 2008-2015 | 16.8 | 238 | 3.7 | |
| Lake Murray | MDN | G | 34.10 | -97.07 | 2008-2015 | 11.0 | 1024.3 | 10.7 | |
| Wichita Mountains NWR | MDN | G | 34.73 | -98.71 | 2008-2015 | 12.4 | 753.2 | 9.2 | |
| Copan | MDN | G | 36.91 | -95.88 | 2008-2015 | 11.7 | 1075.5 | 12.4 | |
| Yellowstone | MDN | G | 44.92 | -110.42 | 2008-2015 | 12.1 | 442.8 | 5.4 | |
| Mt. Waliguan | GMOS | G | 36.29 | 100.90 | 2012–2014 | 5.3 | 94.8 | 1.0 | Sprovieri et al., 2017 |
| Bayinbuluk | | G | 42.90 | 83.72 | 2013/12-2014/12 | 7.7 | 260 | 2.0 | Fu et al., 2016a |
| Nam Co | | G | 30.78 | 90.99 | 2009-2011 | 4.8 | 365 | 1.8 | Huang et al., 2012b |
| Col Margherita | GMOS | G | 46.37 | 11.79 | 2014 | 7.8 | 559.5 | 4.4 | Sprovieri et al., 2017 |
| Hell's Gate National Park | | G | -0.89 | 36.31 | 2009/4-2010/5 | 24.29 | 691 | 16.8 | Wetang'ula, 2011 |
| Glennies Creek | | G | -32.45 | 151.10 | 2006/06-2007/12 | 7.1 | 4102 | 29.1 | Dutt et al., 2009 |
| Everglades N.R.P. | MDN | PW | 26.66 | -80.40 | | 13.6 | 1472.3 | 20.0 | |
| Celestún | GMOS | PW | 20.86 | -90.38 | 2012–2013 | 8.1 | 297.1 | 2.4 | Sprovieri et al., 2017 |
| Bettles | MDN | S | 66.91 | -151.68 | 2008-2015 | 7.6 | 342.135 | 2.5 | |
| Pensacola | MDN | S | 30.55 | -87.38 | 2008-2015 | 9.85 | 1830.6 | 17.8 | |

| Site Name | Network | LUC | Latitude | Longitude | Period | Conc. (ng/L) | Prec. (mm) | Flux (µg m$^{-2}$ yr$^{-1}$) | References |
|---|---|---|---|---|---|---|---|---|---|
| Western Broward County | MDN | S | 26.17 | -80.82 | 2008-2015 | 14.17 | 1292.9 | 18.2 | |
| Indiana Dunes | MDN | S | 41.63 | -87.09 | 2008-2015 | 10.3 | 1068.6 | 10.5 | |
| Stilwell | MDN | S | 35.75 | -94.67 | 2008-2015 | 10.5 | 1256.8 | 12.8 | |
| Yinzidu | | S | 26.57 | 106.12 | 2005/05-2006/09 | 35.7 | 1068 | 38.1 | Guo et al., 2008 |
| Wujiangdu | | S | 27.32 | 106.77 | 2005/05-2006/11 | 57.1 | 693 | 39.6 | Guo et al., 2008 |
| Amsterdam Island | GMOS | S | -37.80 | 77.55 | 2013–2014 | 2.1 | 848.7 | 1.8 | Sprovieri et al., 2017 |
| Delta Elementary | MDN | WS | 30.79 | -87.85 | 2008-2015 | 8.93 | 1891.66 | 16.9 | |
| Centreville | MDN | WS | 32.90 | -87.25 | 2008-2015 | 9.1 | 1522.3 | 12.3 | |
| Bay Road | MDN | WS | 30.47 | -88.14 | 2008-2015 | 8.9 | 1949.98 | 17.4 | |
| Molas Pass | MDN | WS | 37.75 | -107.69 | 2008-2015 | 14.0142 | 884.21 | 12.8 | |
| Okefenokee | MDN | WS | 30.74 | -82.13 | 2008-2015 | 9.85 | 1298.8 | 12.7 | |
| Yorkville | MDN | WS | 33.93 | -85.05 | 2008-2015 | 8.95 | 1362.8 | 12.1 | |
| Hammond | MDN | WS | 30.50 | -90.38 | 2008-2015 | 10.1 | 1586.5 | 16.0 | |
| Smithsonian | MDN | WS | 38.89 | -76.56 | 2008-2015 | 6.9 | 1024.1 | 7.0 | |
| Seney | MDN | WS | 46.29 | -85.95 | 2008-2015 | 8.4 | 790.8 | 6.6 | |
| Leech Lake | MDN | WS | 47.16 | -94.15 | 2008-2015 | 7.4 | 566.2 | 4.3 | |
| Camp Ripley | MDN | WS | 46.25 | -94.50 | 2008-2015 | 9.6 | 792.2 | 7.8 | |
| Grand Bay NERR | MDN | WS | 30.43 | -88.43 | 2008-2015 | 10.3 | 1704.6 | 17.3 | |
| Glacier National Park-Fire W.S. | MDN | WS | 48.51 | -114.00 | 2008-2015 | 6.4 | 889 | 5.8 | |
| Stephenville | MDN | WS | 48.56 | -58.57 | 2008-2015 | 5.2 | 1258.1 | 6.5 | |
| Athens Super Site | MDN | WS | 39.31 | -82.12 | 2008-2015 | 8.2 | 1026.6 | 8.3 | |
| McGee Creek | MDN | WS | 34.32 | -95.89 | 2008-2015 | 10.7 | 1152.9 | 12.1 | |
| Waynesburg | MDN | WS | 39.82 | -80.29 | 2008-2015 | 10.1 | 953.4 | 9.4 | |

| Site Name | Network | LUC | Latitude | Longitude | Period | Conc. (ng/L) | Prec. (mm) | Flux (μg m⁻² yr⁻¹) | References |
|---|---|---|---|---|---|---|---|---|---|
| Hills Creek State Park | MDN | WS | 41.80 | -77.19 | 2008-2015 | 7.2 | 870.4 | 6.2 | |
| Savannah River | MDN | WS | 33.25 | -81.65 | 2008-2015 | 9.3 | 1216 | 11.3 | |
| Bredkälen | EMEP | WS | 63.85 | 15.33 | 2009 | | 412 | 1.3 | Bieser et al., 2014 |
| Pallas | GMOS | WS | 68.00 | 24.24 | 2011–2014 | 6.1 | 340.8 | 2.1 | Sprovieri et al., 2017 |
| HD01 | | WS | 21.16 | -98.37 | 2003/9-2005/12 | 8.2 | 1149 | 9.4 | Hansen and Gay, 2013 |
| Caballo | MDN | OS | 33.06 | -107.29 | 2008-2015 | 14.2 | 372.9 | 5.3 | |
| Birmingham | MDN | U | 33.55 | -86.81 | 2008-2015 | 13.2 | 1474.3 | 19.5 | |
| Fort Collins | MDN | U | 40.59 | -105.14 | 2008-2015 | 11.594 | 508 | 5.9 | |
| Chassahowitzka | MDN | U | 28.75 | -82.56 | 2008-2015 | 11.67 | 1382.8 | 14.7 | |
| Lake Charles | MDN | U | 30.17 | -93.17 | 2008-2015 | 9.5 | 1785.9 | 17.0 | |
| Beltsville | MDN | U | 39.03 | -76.82 | 2008-2015 | 7.1 | 1197.8 | 6.9 | |
| Blaine | MDN | U | 45.14 | -93.22 | 2008-2015 | 11.5 | 782.4 | 9.0 | |
| University Research Farm | MDN | U | 36.07 | -79.73 | 2008-2015 | 6.3 | 1194 | 7.5 | |
| New Brunswick | MDN | U | 40.47 | -74.42 | 2008-2015 | 6.6 | 1134.4 | 7.4 | |
| Bronx | MDN | U | 40.87 | -73.88 | 2008-2015 | 6.9 | 1217.8 | 8.4 | |
| Rochester | MDN | U | 43.15 | -77.55 | 2008-2015 | 7.6 | 961.6 | 7.2 | |
| Beaverton | MDN | U | 45.47 | -122.82 | 2008-2015 | 4.9 | 1151.4 | 5.6 | |
| Salt Lake City | MDN | U | 40.71 | -111.96 | 2008-2015 | 18.6 | 372.7 | 6.9 | |
| Seattle/NOAA | MDN | U | 47.68 | -122.26 | 2008-2015 | 8.1 | 930.5 | 6.9 | |
| Milwaukee | MDN | U | 43.08 | -87.88 | 2008-2015 | 9.9 | 938.1 | 9.3 | |
| Toronto | | U | 43.67 | -79.40 | 2005/06-2008/03 | 15.3 | 1425 | 21.8 | Zhang et al., 2012 |
| Nanjing | | U | 32.05 | 118.78 | 2011/06-2012/02 | 52.9 | 1068 | 56.5 | Zhu et al., 2014 |
| Guiyang | | U | 26.57 | 106.72 | 2012/09-2013/08 | 11 | 1145 | 12.6 | Fu et al., 2016a |

| Site Name | Network | LUC | Latitude | Longitude | Period | Conc. (ng/L) | Prec. (mm) | Flux ($\mu g\ m^{-2}\ yr^{-1}$) | References |
|---|---|---|---|---|---|---|---|---|---|
| Lhasa | | U | 29.63 | 91.01 | 2010 | 24.8 | 331 | 8.2 | Huang et al., 2013 |
| Qingdao | | U | 36.16 | 120.34 | 2016/03-2017/2 | 13.6 | 419 | 5.7 | Chen et al., 2018 |
| Hongwen,Xiamen | | U | 24.48 | 118.16 | 2012/06-2013/05 | 12.5 | 1024 | 12.8 | Xu et al., 2014 |
| Gulangyu,Xiamen | | U | 24.45 | 118.07 | 2012/06-2013/05 | 11.4 | 1001 | 11.4 | Xu et al., 2014 |
| Ningbo | | U | 29.86 | 121.55 | 2007-2008 | 17.1 | 1488.2 | 25.8 | Nguyen et al., 2016 |
| Weinan,Shanxi | | U | 34.48 | 109.48 | 2014/12-2015/12 | 7.4 | 246 | 1.8 | Lu and Liu, 2018 |
| Seoul | | U | 37.51 | 127.00 | | 13.38 | 1300 | 17.4 | Seo et al., 2012 |
| Minamata Bay | | U | 32.21 | 130.41 | 2009/09-2010/08 | 5.9 | 2322 | 13.7 | Marumoto and Matsuyama, 2014 |
| Poznań city | | U | 52.42 | 16.94 | 2013/4-2014/10 | 6.96 | 377 | 2.6 | Siudek et al., 2016 |
| Pretoria | | U | -25.73 | 28.27 | 2007-2010 | 15.8 | 1287 | 20.3 | Gichuki and Manson, 2013 |
| North Ryde | | U | -33.77 | 151.12 | 2006/06-2007/13 | 4.4 | 4139 | 18.2 | Dutt et al., 2009 |
| Nome | MDN | W | 64.51 | -165.40 | 2008-2015 | 6.2 | 379.97 | 2.3 | |
| Yurok Tribe-Requa | MDN | W | 41.56 | -124.09 | 2008-2015 | 4.4 | 1643.78 | 7.0 | |
| Sapelo Island | MDN | W | 31.40 | -81.28 | 2008-2015 | 8.9 | 1089.55 | 9.7 | |
| Casco Bay-Wolfes Neck Farm | MDN | W | 43.83 | -70.06 | 2008-2015 | 5.1 | 1282.6 | 6.5 | |
| Cedar Beach-Southold | MDN | W | 41.03 | -72.39 | 2008-2015 | 4.9 | 982.7 | 4.8 | |
| South Bass Island | MDN | W | 41.66 | -82.83 | 2008-2015 | 8.9 | 796.8 | 7.1 | |
| Erie | MDN | W | 42.16 | -80.11 | 2008-2015 | 8.6 | 1085.2 | 9.4 | |
| Cape Romain | MDN | W | 32.94 | -79.66 | 2008-2015 | 7.6 | 1243.8 | 9.4 | |
| Bermuda | | W | 32.26 | -64.88 | 2008/07-2009/12 | 4.7 | 1531 | 7.2 | Gichuki and Mason, 2014 |
| Xiamen | | W | 24.60 | 118.31 | 2013/07-2014/02 | 26.6 | 1143 | 30.4 | Wu, 2014 |
| Chengshantou | | W | 37.38 | 122.68 | 2007-2008 | 18.2 | 427.7 | 7.8 | Nguyen et al., 2016 |

| Site Name | Network | LUC | Latitude | Longitude | Period | Conc. (ng/L) | Prec. (mm) | Flux (µg m$^{-2}$ yr$^{-1}$) | References |
|---|---|---|---|---|---|---|---|---|---|
| Pengjiayu | | W | 25.63 | 122.07 | 2009 | 8.85 | 1348 | 10.2 | Sheu and Lin, 2013 |
| Koksijde | EMEP | W | 51.45 | 3.30 | 2009 | | 650 | 2.7 | Bieser et al., 2014 |
| Westerland | EMEP | W | 54.12 | 8.30 | 2009 | | 740 | 3.9 | Bieser et al., 2014 |
| Niembro | EMEP | W | 43.43 | -4.85 | 2009 | | 1028 | 1.8 | Bieser et al., 2014 |
| Råö | GMOS | W | 57.39 | 11.91 | 2011–2014 | 9.4 | 603.8 | 5.7 | Sprovieri et al., 2017 |
| Mace Head | GMOS | W | 53.33 | -9.91 | 2012–2014 | 5.7 | 688.6 | 3.3 | Sprovieri et al., 2017 |
| Listvyanka | GMOS | W | 51.85 | 104.89 | 2012–2013 | 6.2 | 32.5 | 0.2 | Sprovieri et al., 2017 |
| Sisal | GMOS | W | 21.16 | -90.05 | 2013–2014 | 8.9 | 691.1 | 7.0 | Sprovieri et al., 2017 |
| OA02 | | W | 15.67 | -96.50 | 2003/9-2005/13 | 7.9 | 894 | 7.1 | Hansen and Gay, 2013 |
| Cape Point | GMOS | W | -34.35 | 18.49 | 2011–2015 | 8.3 | 237.2 | 2.1 | Sprovieri et al., 2017 |
| Cape Grim | GMOS | W | -40.68 | 144.69 | 2013–2015 | 5.7 | 605.1 | 3.3 | Sprovieri et al., 2017 |

Table S2: Litterfall Hg deposition distribution along different forest type (DB: Deciduous Broadleaf Forests; DN: Deciduous Needleleaf Forests; EB: Evergreen Broadleaf Forests; EN: Evergreen Needleleaf Forests; MIX: Mixed Forests; G: Grasslands)

| Site name | Forest type | Latitude | Longitude | Elevation (m) | Period | Flux (µg m$^{-2}$ yr$^{-1}$) | Conc. (ng/L) | Reference |
|---|---|---|---|---|---|---|---|---|
| Mt. Damei | DB | 29.63 | 121.57 | 550 | 2012/08-2013/07 | 23.10 | 42.30 | Fu et al., 2016 |
| Mt. Dongling | DB | 40.00 | 115.43 | 1100 | 2015/09-2015/11 | 15.60 | 48.70 | Zhou et al., 2017 |
| Roush Lake | DB | 40.84 | -85.46 | 244 | 2007-2009 2012-2014 | 13.90 | 39.93 | Risch et al., 2017 |
| Clifty Falls | DB | 38.76 | -85.42 | 256 | 2007-2009 2012-2014 | 16.03 | 48.73 | Risch et al., 2017 |
| SW Agr. Center | DB | 38.74 | -87.49 | 134 | 2015-2016 | 17.30 | 45.45 | Risch and Kenski, 2018 |
| Fort Harrison | DB | 39.86 | -86.02 | 260 | 2007-2009 2012-2014 | 17.37 | 46.27 | Risch et al., 2017 |
| Indiana Dunes | DB | 41.63 | -87.09 | 208 | 2007-2009 2012-2014 | 17.73 | 42.32 | Risch et al., 2017 |
| Mammoth Cave | DB | 37.13 | -86.15 | 236 | 2007-2009 2012-2014 | 12.60 | 40.40 | Risch et al., 2017 |
| Piney Reservoir | DB | 39.71 | -79.01 | 769 | 2007-2009 2012-2014 | 15.30 | 54.80 | Risch et al., 2017 |
| Marcell | DB | 47.53 | -93.47 | 431 | 2007-2009 2012-2014 | 3.58 | 25.80 | Risch et al., 2017 |
| Blaine | DB | 45.14 | -93.22 | 275 | 2007-2009 2012-2014 | 7.80 | 37.15 | Risch et al., 2017 |
| Mingo | DB | 36.97 | -90.14 | 105 | 2007-2009 2012-2014 | 13.27 | 39.80 | Risch et al., 2017 |
| Huntington Wildlife | DB | 43.97 | -74.22 | 500 | 2007-2009 2012-2014 | 11.25 | 45.10 | Risch et al., 2017 |
| Biscuit Brook | DB | 41.99 | -74.50 | 634 | 2007-2009 2012-2014 | 12.20 | 53.45 | Risch et al., 2017 |
| Athens | DB | 39.31 | -82.12 | 275 | 2007-2009 2012-2014 | 18.83 | 50.10 | Risch et al., 2017 |
| Allegheny Portage | DB | 40.46 | -78.56 | 739 | 2007-2009 2012-2014 | 13.38 | 45.46 | Risch et al., 2017 |
| Leading Ridge | DB | 40.66 | -77.94 | 287 | 2007-2009 2012-2014 | 9.60 | 44.24 | Risch et al., 2017 |
| Cape Romaine | DB | 32.94 | -79.66 | 1 | 2007-2009 2012-2014 | 14.87 | 40.47 | Risch et al., 2017 |
| Smoky Mountains | DB | 35.66 | -83.59 | 640 | 2007-2009 2012-2014 | 7.75 | 36.00 | Risch et al., 2017 |

| Site name | Forest type | Latitude | Longitude | Elevation (m) | Period | Flux ($\mu g\ m^{-2}\ yr^{-1}$) | Conc. (ng/L) | Reference |
|---|---|---|---|---|---|---|---|---|
| Shenandoah | DB | 38.52 | -78.43 | 1072 | 2007-2009 2012-2014 | 11.25 | 36.55 | Risch et al., 2017 |
| Popple River | DB | 45.80 | -88.40 | 421 | 2007-2009 2012-2014 | 10.00 | 34.30 | Risch et al., 2017 |
| Devil's Lake | DB | 43.44 | -89.68 | 389 | 2007-2009 2012-2014 | 4.60 | 35.47 | Risch et al., 2017 |
| Trout Lake | DB | 46.05 | -89.65 | 509 | 2007-2009 | 7.60 | 35.63 | Risch et al., 2012 |
| Bad River | DB | 46.60 | -90.67 | 625 | 2007-2009 2012-2014 | 7.77 | 29.50 | Risch et al., 2017 |
| Lake Geneva | DB | 42.58 | -88.50 | 288 | 2007-2009 2012-2014 | 14.30 | 49.80 | Risch et al., 2017 |
| Canaan Valley | DB | 39.06 | -79.42 | 988 | 2007-2009 2012-2014 | 9.33 | 39.20 | Risch et al., 2017 |
| Coolidge | DB | 43.541 | -72.754 | 552 | 2008-2009 | 24 | 73.9 | Juillerat et al., 2012 |
| Steam Mill Brook | DB | 44.478 | -72.228 | 649 | 2008-2009 | 18.1 | 81.4 | Juillerat et al., 2012 |
| The Waterworks | DB | 44.171 | -73.078 | 237 | 2008-2009 | 15.4 | 69.1 | Juillerat et al., 2012 |
| Wheaton College Campus | DB | 41.964 | -71.18 | 20 | 2010 | 10 | 33 | Benoit etal ., 2013 |
| Camels Hump | DB | 44.32 | -72.886 | 1244 | 2010 | | | Stankwitz et al., 2012 |
| Sao Francisco do Itabapoana | DB | -21.6 | -41.05 | 20 | 2006 | 24 | 31 | Fragoso et al., 2018 |
| Yangsu-ri | DB | 37.53 | 127.33 | 60 | 2008/08-2010/02 | 2.1 | 50.2 | Han et al., 2016 |
| | | | | | 2005/03-2006/03; 2010-2011; | | | Wang et al., 2009; Luo et al., 2015; |
| Tieshanping | DN | 29.62 | 106.69 | 500 | 2014/04-2015/03; | 102.66 | 104.80 | Zhou et al., 2016 |
| Mt. Ailao | EB | 23.70 | 101.02 | 2450 | 2011-2014 | 75.00 | 58.00 | Wang et al., 2016 |
| Mt. Gongga | EB | 29.58 | 101.93 | 3000 | 2005/05-2006/04 | 35.50 | 35.70 | Fu et al., 2010a |
| Mt. Jinyun | EB | 29.87 | 106.41 | 900 | 2012/03-2013/02 | 43.50 | 104.50 | Ma et al., 2015 |
| Alta Floresta | EB | -9.95 | -56.33 | 300 | 2013 | 44.825 | 55 | Fostier et al., 2015 |
| Candeias de Jamari | EB | -8.73 | -63.47 | 90 | 2014 | 49.715 | 61 | Fostier et al., 2015 |

| Site name | Forest type | Latitude | Longitude | Elevation (m) | Period | Flux ($\mu$g m$^{-2}$ yr$^{-1}$) | Conc. (ng/L) | Reference |
|---|---|---|---|---|---|---|---|---|
| Candeias de Jamari | EB | -8.63 | -63.35 | 90 | 2014 | 46.455 | 57 | Fostier et al., 2015 |
| Rio de Janeiro | EB | -22.88 | -43.88 | 250 | 2005-2006 | 184 | 237 | Teixeira et al., 2012 |
| Serra da Mantiqueira | EB | -22.302 | -44.452 | 2000 | 2009-2011 | 34.6 | 57 | Teixeira et al., 2017 |
| Taquara | EB | -22.502 | -42.856 | 72 | 2013 | 123.12 | 240 | Buch et al., 2015 |
| Três Picos State Park | EB | -22.598 | -43.239 | 74 | 2013 | 20 | 40 | Buch et al., 2015 |
| Linzhi | EN | 29.85 | 94.70 | 3200 | 2008/08-2008/09 | 4.20 | 12.60 | Gong et al., 2014 |
| Huitong | EN | 26.83 | 109.75 | 400 | 2013-2014 | 33.60 | 176.10 | Luo et al., 2015 |
| Qianyanzhou, Jiangxi | EN | 26.75 | 115.07 | 105 | 2013-2014 | 21.40 | 42.90 | Luo et al., 2015 |
| Great Smoky Mountains | EN | 35.556 | -83.5 | 1308 | 2008-2009 | 17.93 | 50.035 | Fisher and Wolfe, 2012 |
| Thompson | EN | 47.384 | -121.94 | 220 | 2008 | 8.7 | | Obrist et al., 2012 |
| Truckee | EN | 39.43 | -120 | 1700 | 2007 | 20 | 9.238 | Obrist et al., 2009 |
| Lesní potok catchment | EN | 49.967 | 13.683 | 448 | 2008-2012 | 18.5 | | Navrátil et al., 2014 |
| Langtjern | EN | 60.13 | 10.783 | 500 | 2004-2005 | 10.3 | 38 | Larssen et al., 2008 |
| Mt. Changbai | MIX | 42.40 | 128.11 | 736 | 2013/10-/2013/10 | 22.80 | 47.00 | Wan et al., 2009; Fu et al., 2016 |
| Mt. Leigong | MIX | 26.39 | 108.20 | 2176 | 2008/05-2009/04 | 39.50 | 91.10 | Fu et al., 2010b; 2016 |
| Mt. Simian | MIX | 28.58 | 106.37 | 1394 | 2012/03-2013/02 | 42.89 | 106.70 | Ma et al., 2016 |
| LuChongGuan | MIX | 26.37 | 106.72 | 1360 | 2005/01-2006/01 | | | Wang et al., 2009 |
| Okefenokee | MIX | 30.74 | -82.13 | 45 | 2007-2009 2012-2014 | 4.50 | 31.54 | Risch et al., 2017 |
| Beltsville | MIX | 39.03 | -76.82 | 46 | 2007-2009 2012-2014 | 14.62 | 42.24 | Risch et al., 2017 |
| Seney | MIX | 46.29 | -85.95 | 220 | 2007-2009 2012-2014 | 6.26 | 41.80 | Risch et al., 2017 |
| Presque Isle | MIX | 42.16 | -80.11 | 177 | 2007-2009 2012-2014 | 7.55 | 28.85 | Risch et al., 2017 |
| Whiteface Mountain | MIX | 44.37 | -73.9 | 1483 | 2015 | 7 | 48.667 | Gerson et al., 2017 |

Table S3: Speciated Hg dry deposition distribution along different land use type (B: Barren; C/N: Cropland/Natural Vegetation mosaics; C: Croplands; DB: Deciduous Broadleaf Forests; DN: Deciduous Needleleaf Forests; EB: Evergreen Broadleaf Forests; EN: Evergreen Needleleaf Forests; MIX: Mixed Forests; G: Grasslands; PW: Permenant Wetlands; S: Savannas; WS: Woody Savannas; OS: Open Shrublands; Urban and Built-up Lands; W: Water.)

| Site name | LUC | Latitude | Longitude | Period | Flux ($\mu$g m$^{-2}$ yr$^{-1}$) | | | | References | Methods |
|---|---|---|---|---|---|---|---|---|---|---|
| | | | | | TM | GEM | GOM | PBM | | |
| North District Ranger | B | 38.12 | -119.81 | 2010/06-2010/12; 2011/03-2011/10; 2012/02-2012/06 | | | 2.63 | | Wright et al., 2014 | CEM |
| Mauna Loa | B | 19.54 | -155.58 | 2011-2014 | 69.80 | 4.28 | 33.75 | 10.00 | Zhang et al., 2016b | Estimated |
| Cortez mine | B | 40.20 | -116.10 | 2008-2009 | | -5.26 | | | Miller et al., 2011 | DFC |
| Twin Creeks | B | 41.20 | -117.10 | 2008-2009 | | -32.41 | | | Miller et al., 2011 | DFC |
| Beiluhe | B | 34.83 | 92.94 | 2013/06-2014/06 | | -25.05 | | | Ci et al., 2016 | DFC |
| Longview | C/N | 32.38 | -94.71 | 2011/09-2012/09 | | | 1.49 | | Sather et al., 2014 | CEM |
| Corsicana | C/N | 35.75 | -94.67 | 2011/09-2012/09 | | | 3.00 | | Sather et al., 2014 | CEM |
| Karnack | C/N | 32.67 | -94.16 | 2011/09-2012/09 | | | 1.31 | | Sather et al., 2014 | CEM |
| Westing Park | C/N | 24.25 | 120.80 | 2011–2012 | | | | 11.04 | Fang et al., 2012b | DDP |
| Taichung airport | C/N | 24.25 | 120.60 | 2011–2012 | | | | 12.09 | Fang et al., 2012b | DDP |
| Lhasa branch | C/N | 29.63 | 91.15 | 2013/04-2014/08 | | | | 35.30 | Huang et al., 2016 | TSP/Estimated |
| Nanjing | C/N | 32.05 | 118.78 | 2011/06-2012/02 | | | | 62.60 | J. Zhu et al., 2014 | Anderson/Estimated |
| Kathmandu Valley | C/N | 27.69 | 85.40 | 2013/04-2014/04 | | | | 134.00 | Guo et al., 2017 | TSP/Estimated |
| Waldhof | C/N | 52.80 | 10.75 | 2009 | | 4.10 | 3.30 | 1.65 | Bieser et al., 2014 | Estimated |
| Yarner Wood | C/N | 50.59 | -3.70 | 2009 | | 3.80 | 4.10 | 2.05 | Bieser et al., 2014 | Estimated |
| Elkhorn Slough | C | 36.81 | -121.78 | 2010-2011 | 11.25 | 10.90 | 0.10 | 0.20 | Zhang et al., 2016b | Estimated |
| Antelope Island | C | 41.09 | -112.12 | 2010 | | 12.10 | 1.70 | | Zhang et al., 2016b | Estimated |
| Horicon | C | 43.47 | -88.62 | 2011-2014 | 14.28 | 12.95 | 0.60 | 0.73 | Zhang et al., 2016b | Estimated |
| Presque Isle | C | 46.70 | -68.03 | 2014 | 15.40 | 14.60 | 0.50 | 0.30 | Zhang et al., 2016b | Estimated |
| Elora | C | 43.65 | -80.42 | 2006/11-2007/8 | | -55.19 | | | Baya and Heyst., 2010 | AGM |
| Yucheng | C | 36.95 | 116.60 | 2012-2013 | | -62.20 | | | Sommar et al., 2016b | REA |

| Site name | LUC | Latitude | Longitude | Period | Flux (μg m$^{-2}$ yr$^{-1}$) | | | | References | Methods |
|---|---|---|---|---|---|---|---|---|---|---|
| | | | | | TM | GEM | GOM | PBM | | |
| Zingst | C | 54.43 | 12.73 | 2009 | | 0.80 | 2.30 | 1.15 | Bieser et al., 2014 | Estimated |
| Heigham Holmes | C | 52.72 | -1.62 | 2009 | | 1.50 | 2.40 | 1.20 | Bieser et al., 2014 | Estimated |
| Auchencorth Moss | C | 55.78 | -3.23 | 2009 | | 2.10 | 2.10 | 1.05 | Bieser et al., 2014 | Estimated |
| Banchory | C | 57.07 | -2.53 | 2009 | | 2.10 | 2.10 | 1.05 | Bieser et al., 2014 | Estimated |
| Kollumerwaard | C | 52.30 | 4.50 | 2009 | | 3.00 | 3.40 | 1.70 | Bieser et al., 2014 | Estimated |
| Birkenes | C | 58.38 | 8.25 | 2009 | | 4.10 | 2.70 | 1.35 | Bieser et al., 2014 | Estimated |
| Piney Creek | DB | 39.71 | -79.01 | 2009/09-2010/10 | 14.40 | 3.50 | 3.20 | | Castro et al., 2012 | CEM |
| Canaan Valley | DB | 39.06 | -79.42 | 2009 2011 | | 3.05 | | | Zhang et al., 2016b | Estimated |
| Huntington | DB | 43.97 | -74.22 | 2009-2012 2014 | 9.14 | 8.76 | 3.24 | 0.42 | Zhang et al., 2016b | Estimated |
| Underhill | DB | 44.53 | -72.87 | 2009-2014 | 9.17 | 8.38 | 0.33 | 0.47 | Zhang et al., 2016b | Estimated |
| Shenandoah | DB | 38.51 | -78.44 | 2008-2009 | | 9.92 | | | Converse et al., 2010 | MBR |
| Mt. Leigong | DB | 26.39 | 108.20 | 2008/05-2009/05 | 43.90 | | | | Fu et al., 2010a | Forest |
| Yangsu-ri | DB | 37.53 | 127.33 | 2008/08-2010/02 | 9.90 | | 4.78 | 4.37 | Han et al., 2016 | KSS |
| Tieshanping | DN | 29.62 | 106.69 | 2005/03-2006/03 | 262.30 | | | | Wang et al., 2009 | Forest |
| Chalk Mountain | EB | 37.16 | -122.29 | 2012/06-2012/10 | | | 4.38 | | Huang and Gustin, 2015a | CEM |
| Mt. Ailao | EB | 24.54 | 101.03 | 2011-2014 | 95.10 | | | | Wang et al., 2016 | Forest |
| Mt. Gongga | EB | 29.58 | 101.93 | 2005/05-2006/04 | 66.50 | | | | Fu et al., 2010b | Forest |
| Mt. Jinyun | EB | 29.69 | 106.30 | 2012/03-2013/02 | 49.40 | | | | Ma et al., 2013; 2015 | Forest |
| Lower Kaweah | EN | 36.57 | -118.78 | 2010/06-2010/12; 2011/03-2011/10; 2012/02-2012/06 | | | 7.88 | | Wright et al., 2014 | CEM |
| E.L.A | EN | 49.66 | -92.23 | 2008–2009 | 16.30 | 15.60 | 0.49 | 0.25 | L. Zhang et al., 2012 | Estimated |
| Qianyanzhou | EN | 26.75 | 115.07 | 2013/12-2014/11 | 52.50 | -40.21 | | | Luo et al., 2016 | AGM |

| Site name | LUC | Latitude | Longitude | Period | Flux ($\mu g\ m^{-2}\ yr^{-1}$) | | | | References | Methods |
|---|---|---|---|---|---|---|---|---|---|---|
| | | | | | TM | GEM | GOM | PBM | | |
| Huitong | EN | 26.83 | 109.75 | 2013/12-2014/12 | 45.50 | -17.78 | | | Luo et al., 2015 | Forest |
| Schmücke | EN | 50.65 | 10.77 | 2009 | | 4.10 | 4.00 | 2.00 | Bieser et al., 2014 | Estimated |
| Iskrba | EN | 45.57 | 14.87 | 2009 | | 0.80 | 5.40 | 2.70 | Bieser et al., 2014 | Estimated |
| Thompson Farm | MIX | 43.11 | -70.95 | 2009-2011 | 7.70 | 6.67 | 1.94 | 1.10 | Zhang et al., 2016b | Estimated |
| Kejimkujik | MIX | 44.43 | -65.21 | 2009-2014 | 22.02 | 21.24 | 0.15 | 0.63 | Zhang et al., 2016b | Estimated |
| Mt. Simian | MIX | 28.62 | 106.40 | 2012/03-2013/02 | 59.61 | | | | Ma et al., 2016 | Forest |
| Pinet | MIX | 42.87 | 1.97 | 2010-2013 | | 38.00 | 0.40 | 0.20 | Enrico et al., 2016b | Estimated |
| Schauinsland | MIX | 47.90 | 7.90 | 2009 | | 4.30 | 3.00 | 1.50 | Bieser et al., 2014 | Estimated |
| Vavihill | MIX | 56.02 | 13.15 | 2009 | | 3.70 | 3.20 | 1.60 | Bieser et al., 2014 | Estimated |
| Fort Parker | G | 31.61 | -96.55 | 2011/09-2012/09 | | | 1.14 | | Sather et al., 2014 | CEM |
| Mesa Verde | G | 37.20 | -108.49 | 2010/08-2011/08 | | | 8.58 | | Sather et al., 2013 | CEM |
| Valles Caldera | G | 35.86 | -106.52 | 2009/08-2011/08 | | | 5.33 | | Sather et al., 2013 | CEM |
| Navajo Lake | G | 36.81 | -107.65 | 2009/08-2011/08 | | | 5.30 | | Sather et al., 2013 | CEM |
| Paradise Valley | G | 41.50 | -117.50 | 2005/03-2005/11; 2006/12-2007/2 | | | 5.10 | | Lyman et al., 2007; Huang and Guatin, 2015a | CEM |
| Gibbs Ranch | G | 41.57 | -115.21 | 2005/03-2005/11; 2006/12-2007/2 | | | 4.54 | | Lyman et al., 2007 | CEM |
| Angel Peak | G | 36.32 | -115.57 | 2013/05-2013/12 | | | 20.15 | | Huang and Gustin, 2015a | CEM |
| Crooked Creek | G | 37.50 | -118.17 | 2012/06-2012/10; 2013/06-2013/09 | | | 10.51 | | Huang and Gustin, 2015a | CEM |
| Echo Peak | G | 37.22 | -116.33 | 2012/06-2012/11; 2013/02-2013/011 | | | 24.53 | | Huang and Gustin, 2015a | CEM |

| Site name | LUC | Latitude | Longitude | Period | Flux (µg m$^{-2}$ yr$^{-1}$) | | | | References | Methods |
|---|---|---|---|---|---|---|---|---|---|---|
| | | | | | TM | GEM | GOM | PBM | | |
| Great Basin | G | 39.01 | -114.22 | 2012/06-2012/10 | | | 22.78 | | Huang and Gustin, 2015a | CEM |
| Peavine Peak | G | 39.59 | -119.93 | 2013/04-2013/10 | | | 25.40 | | Huang and Gustin, 2015a | CEM |
| Cathedral Gorge | G | 37.51 | -114.63 | 2012/09-2012/12; 2013/01-2013/12 | | | 6.13 | | Huang and Gustin, 2015a | CEM |
| Lehman | G | 39.01 | -114.22 | 2010/06-2010/12; 2011/03-2011/10; 2012/02-2012/06 | | | 16.06 | | Wright et al., 2014 | CEM |
| Substation | OS | 36.80 | -108.48 | 2009/08-2011/08 | | | 6.87 | | Sather et al., 2013 | CEM |
| Berlin Ichthyosaur | OS | 38.57 | -117.78 | 2012/06-2012/9; 2013/06-2013/09 | | | 19.27 | | Huang and Gustin, 2015a | CEM |
| Pahrump | OS | 36.34 | -116.03 | 2013/05-2013/12 | | | 8.76 | | Huang and Gustin, 2015a | CEM |
| Brigantine | PW | 39.46 | -74.45 | 2009 | | 17.80 | | | Zhang et al., 2016b | Estimated |
| Stilwell | S | 35.75 | -94.67 | 2009-2013 | 7.26 | 6.46 | 1.73 | 0.32 | Sather et al., 2013; Zhang et al., 2016b | Estimated |
| OLF | S | 30.55 | -87.38 | 2009-2014 | 8.00 | 7.40 | 1.87 | 0.20 | Zhang et al., 2016b; Peterson et al., 2012; Huang et al., 2014; 2017 | Estimated |
| Molas Pass | WS | 37.75 | -107.69 | 2009/08-2011/08 | | | 3.00 | | Sather et al., 2013 | CEM |
| Turtleback Dome | WS | 37.71 | -119.71 | 2010/06-2010/12; 2011/03-2011/10; 2012/02-2012/06 | | | 10.22 | | Wright et al., 2014 | CEM |
| Grand Bay NERR | WS | 30.43 | -88.43 | 2009-2014 | 16.78 | 13.55 | 2.38 | 0.85 | Zhang et al., 2016b | Estimated |

| Site name | LUC | Latitude | Longitude | Period | Flux (µg m$^{-2}$ yr$^{-1}$) | | | | References | Methods |
|---|---|---|---|---|---|---|---|---|---|---|
| | | | | | TM | GEM | GOM | PBM | | |
| Yorkville | WS | 33.93 | -85.05 | 2009-2014 | 11.55 | 10.65 | 1.92 | 0.35 | Zhang et al., 2016b; Lyman et al., 2009; Weiss-Penzias et al., 2011 | Estimated |
| Athens | WS | 39.31 | -82.12 | 2009-2011 2014 | 4.63 | 2.50 | 1.45 | 0.70 | Zhang et al., 2016b | Estimated |
| Pallas (Matorova) | WS | 68.00 | 24.23 | 2009 | | 3.00 | 3.20 | 1.60 | Bieser et al., 2014 | Estimated |
| Bredkälen | WS | 63.85 | 15.33 | 2009 | | 3.40 | 2.70 | 1.35 | Bieser et al., 2014 | Estimated |
| Farmington Airport | U | 36.74 | -108.23 | 2009/08-2011/08 | | | 5.67 | | Sather et al., 2013 | CEM |
| D.R.I. | U | 39.57 | -119.80 | 2008/05-2009/02 | | | 9.98 | | Lyman et al., 2007 | CEM |
| Reno | U | 39.51 | -119.72 | 2006/10-2008/10 | | | 6.80 | | Lyman et al., 2009 | CEM |
| Tampa | U | 27.91 | -82.38 | 2009/07-2010/07 | | | 2.95 | | Peterson et al., 2012; Gustin et al., 2012 | CEM |
| Davie | U | 26.07 | -80.24 | 2009/07-2010/07 | | | 2.78 | | Peterson et al., 2012; Gustin et al., 2012 | CEM |
| New Brunswick | U | 40.47 | -74.42 | 2008–2009 | 23.90 | 23.00 | 0.90 | | L. Zhang et al., 2012 | Estimated |
| Salt Lake City | U | 40.71 | -111.96 | 2009 | 10.80 | 4.04 | 4.35 | 1.00 | Zhang et al., 2016b | Estimated |
| Birmingham | U | 33.55 | -86.81 | 2009-2014 | 2.35 | -1.45 | 5.33 | 0.58 | Zhang et al., 2016b | Estimated |
| Beltsville | U | 39.03 | -76.82 | 2009-2014 | 13.47 | 9.43 | 1.33 | 0.72 | Zhang et al., 2016b | Estimated |
| NYC | U | 40.87 | -73.88 | 2011 2012 2014 | 8.90 | 6.57 | 1.37 | 0.63 | Zhang et al., 2016b | Estimated |
| Qingdao | U | 36.27 | 120.50 | 2008-2011 | | | | 16.06 | Zhang et al., 2015 | TSP/estimated |
| Hungkuang | U | 24.22 | 120.58 | 2010/11-2011/07 | | | | 61.00 | Fang et al., 2012a | MOUDI/Estimated |
| Quanxing | U | 24.14 | 120.49 | 2010/11-2011/07 | | | | 85.00 | Fang et al., 2012a | MOUDI/Estimated |
| South Bass Island | W | 41.66 | -82.83 | 2013 | | 5.20 | | 0.80 | Zhang et al., 2016b | Estimated |

| Site name | LUC | Latitude | Longitude | Period | Flux (μg m$^{-2}$ yr$^{-1}$) | | | | References | Methods |
|---|---|---|---|---|---|---|---|---|---|---|
| | | | | | TM | GEM | GOM | PBM | | |
| The Gironde Estuary | W | 45.16 | -0.72 | 2005-2008 | | -3.50 | | | Castelle et al., 2009 | Estimated |
| Koksijde | W | 51.12 | 2.64 | 2009 | | 3.80 | 3.30 | 1.65 | Bieser et al., 2014 | Estimated |
| Westerland | W | 54.13 | 8.87 | 2009 | | 3.50 | 2.90 | 1.45 | Bieser et al., 2014 | Estimated |
| Niembro | W | 43.43 | -4.85 | 2009 | | 1.20 | 3.50 | 1.75 | Bieser et al., 2014 | Estimated |
| Ráö | W | 57.39 | 11.90 | 2009 | | 2.30 | 3.20 | 1.60 | Bieser et al., 2014 | Estimated |

---

## Author Response (AR2)

**Reply to Comments from Reviewer #1**

The authors should be applauded for their efforts to improve their manuscript. Based on the reply to reviewers' comments/suggestions and the revised manuscript, I believe that the addition of uncertainty analysis improves the scientific contribution. However, some of the major concerns by the three reviewers were not addressed. The revised manuscript is rather long (58 pages) and difficult to follow, hindering its scientific contributions.

*Response:*

We thank the reviewer for the useful comments. We have carefully revised our manuscript according to these comments. The manuscript has been shortened (52 pages). The point-by-point responses are as follows.

*Comment:*

The revised title, "Uncertainties in the observation and simulation of global speciated atmospheric mercury deposition to terrestrial surfaces", may not reflect well the materials presented, i.e. Hg deposition over the land surfaces and the uncertainty associated with measurements and simulations. The authors may want to revise the title or shorten substantially the Hg deposition part.

*Response:*

We thank the reviewer for the valuable comment. We have revised the title to be "Atmospheric mercury deposition over the land surfaces and the associated uncertainties in observations and simulations: a critical review".

*Comment:*

The uncertainty analysis has much room for improvements. For example, time frame should be considered. The uncertainties could be low in estimates of annual precipitation and wet deposition, but could be much higher in event-based cases. As a review article, a range of values should be presented instead of a few values for each item. Overall, the current version is rather descriptive without much in-depth discussion one would expect in a review article.

*Response:*

The uncertainty analysis part has been improved. The time frame has been considered for wet deposition. Please refer to Lines 270–275 in the revised manuscript:

"The measurements of precipitation volume by samplers have non-negligible uncertainties (Wetherbee, 2017). The relative standard deviations (RSDs) of daily and annual precipitation depth measurements in MDN were estimated to be 15 % and 10 %, respectively (Wetherbee et al., 2005). The event-based sampling volume biases of two types of samplers used in APMMN were estimated to be up to 11–18 % (Sheu et al., 2019)."

With the time frame considered, the overall relative uncertainty of the precipitation Hg wet deposition flux was estimated to be ±(15–20) %. Moreover, all the other uncertainties for dry and forest deposition have been updated to uncertainty ranges with different factors considered. Please refer to the relative paragraphs in Section 3 and 4. We have polished the discussion in Section 2, 3 and 4, and deleted some redundant descriptions which have been presented in some previous review articles in details. Some unnecessary figures have been removed.

The uncertainty analysis in this review work is semi-quantitative. The purpose of this study is to identify the most important contributors to the overall uncertainty of global Hg deposition observation and simulation, and recommend crucial research focuses.

**Reply to Comments from Reviewer #2**

The revised version reads better than the ACPD version, but some improvements can be made before it can be accepted for final publication. Below is a list of some specific comments for the authors to consider. Line numbers used here are from the revised manuscript (not the one posted on ACPD)

*Response:*

We thank the reviewer for the useful comments. We have carefully revised our manuscript according to these comments. The point-by-point responses are as follows.

*Comment:*

Line 45: TGM (GEM+GOM) is usually monitored and commonly used in literature, but TM is seldom used.

*Response:*

We have revised this sentence. Please refer to Lines 44–46 in the revised manuscript:

"The sum of GEM and GOM is called total gaseous mercury (TGM), and the sum of GOM and PBM is also known as reactive mercury (RM)."

*Comment:*

Lines 99-101: The Asia–Pacific Mercury Monitoring Network was established under the effort of Taiwan central University. There might be a more appropriate reference for this network from their publications than the one cited here.

*Response:*

Sheu et al. (2019) comprehensively introduced the development, methods, current results and future plan of the Asia-Pacific Mercury Monitoring Network, so we changed the reference here from Obrist et al. (2018) to Sheu et al. (2019). Please refer to Line 100 in the revised manuscript:

"A new Asia–Pacific Mercury Monitoring Network (APMMN) has recently been established (Sheu et al., 2019)."

*Reference: Sheu, G.-R., Gay, D. A., Schmeltz, D., Olson, M., Chang, S.-C., Lin, D.-W., and*

*Nguyen, L. S. P.: A new monitoring effort for Asia: the Asia Pacific Mercury Monitoring Network (APMMN), Atmosphere, 10(9), 481, 2019.*

**Comment:**

Line 107-108: Values in bracket are multiple year range of annual average? Better use this format "Annual average (multiple year range) of Hg wet deposition…"

*Response:*

Yes, the values in brackets are multiple year ranges of the annual averages. We have revised this sentence accordingly. Please refer to Lines 105–107 in the revised manuscript:

"The annual averages (multiple year ranges) of Hg wet deposition in the northern hemisphere, the tropics, and the southern hemisphere were 2.9 (0.2–6.7), 4.7 (2.4–7.0), and 1.9 (0.3–3.3) $\mu g$ $m^{-2}$ $yr^{-1}$, respectively."

**Comment:**

Line 120-122: the second half of the sentence does not explain the first half because GEM is not soluble. You should cite references showing high RM (especially GOM) in Asia.

*Response:*

Travnikov et al. (2017) used four global models to simulate RM concentration and found that the RM concentration in East Asia is significantly higher than in other regions. We have revised this sentence according to the comment and added this reference. Please refer to Line 122 and Line 127 in the revised manuscript:

"Overall, East Asia has the highest wet deposition flux (averagely 16.1 $\mu g$ $m^{-2}$ $yr^{-1}$), especially in the southern part of China where the RM concentration level is relatively high (Fu et al., 2008; Guo et al., 2008; Wang et al., 2009; Fu et al., 2010a; 2010b; Ahn et al., 2011; Huang et al., 2012b; Seo et al., 2012; Huang et al., 2013a; Sheu and Lin, 2013; Marumoto and Matsuyama, 2014; Xu et al., 2014; Zhu et al., 2014; Huang et al., 2015; Zhao et al., 2015; Han et al., 2016; Fu et al., 2016a; Ma et al., 2016; Nguyen et al., 2016; Qin et al., 2016; Sommar et at., 2016; Cheng et al., 2017; Travnikov et al., 2017; Chen et al., 2018; Lu and Liu, 2018)."

*Reference: Travnikov, O., Angot, H., Artaxo, P., Bencardino, M., Bieser, J., D'Amore, F., Dastoor, A., De Simone, F., Diéguez, M. d. C., Dommergue, A., Ebinghaus, R., Feng, X. B.,*

*Gencarelli, C. N., Hedgecock, I. M., Magand, O., Martin, L., Matthias, V., Mashyanov, N., Pirrone, N., Ramachandran, R., Read, K. A., Ryjkov, A., Selin, N. E., Sena, F., Song, S. J., Sprovieri, F., Wip, D., Wängberg, I., and Yang, X.: Multi-model study of mercury dispersion in the atmosphere: atmospheric processes and model evaluation, Atmos. Chem. Phys., 17, 5271–5295, 2017.*

*Comment:*

The paragraph starting on line 142: A substantial revision is needed here. Wet deposition flux mainly depends on precipitation amount, column air concentration, and sometimes long-range transport from in-cloud scavenging, and depends little on underlying surfaces. The logic used in this paragraph is not appropriate. Even if you generated some dependence of wet deposition on the surface type, the conclusion may be applicable at different regions. However, you can indeed compare urban, rural and remote sites (instead of land use or canopy types).

*Response:*

We agree with the reviewer on this point. Since this paper is now focusing on the uncertainties in the observation and simulation of atmospheric Hg deposition, we have deleted this paragraph and made changes in other related contents.

*Comment:*

Section 2.2: Measurement and modeling approaches for quantifying dry deposition of GOM and PBM and air-surface exchange fluxes of GEM, and field studies measuring GOM and PBM dry deposition and mercury in litterfall and throughfall were reviewed in detail by Wright et al. (2016). Measurement and modeling studies of air-surface exchange of GEM were also reviewed by Zhu et al. (2016). In this section, these earlier reviews shroud be first mentioned and then point out what additional knowledge this review is intend to provide. This section is poorly written in my opinion. A synthesis of existing literature is needed, instead of listing results from certain individual publications.

*Response:*

[revised manuscript text omitted]

We have rewritten this section to make it simpler and more readable. Please refer to Lines 169–204 in the revised manuscript:

"Hg deposition in forests is mainly in the forms of litterfall and throughfall. Wright et al. (2016) also made an extensive review of litterfall and throughfall Hg deposition. Wang et al. (2016a) made a comprehensive assessment of the global Hg deposition through litterfall, and found litterfall Hg deposition an important input to terrestrial forest ecosystems (1180±710 Mg yr$^{-1}$). Not many new studies on forest Hg deposition have been reported since then. Therefore, here we only briefly introduce the spatial distribution of forest Hg deposition. South America was estimated to bear the highest litterfall Hg deposition ($65.8\pm57.5$ µg m$^{-2}$ yr$^{-1}$) around the world (Teixeira et al., 2012; Buch et al., 2015; Fostier et al., 2015; Teixeira et al., 2017; Fragoso et al., 2018; Shen et al., 2019). There have been numerous forest Hg deposition studies in the recent decade in East Asia with the second highest average litterfall Hg deposition flux ($35.5\pm27.7$ µg m$^{-2}$ yr$^{-1}$) (Wan et al., 2009; Wang et al., 2009; Fu et al., 2010a; Fu et al., 2010b; Gong et al., 2014; Luo et al., 2016; Ma et al., 2015; Han et al., 2016; Fu et al., 2016a; Ma et al., 2016; Wang et al., 2016b; Zhou et al., 2016; Zhou et al., 2017). Lower levels of litterfall Hg deposition fluxes were found in North America ($12.3\pm4.9$ µg m$^{-2}$ yr$^{-1}$) and Europe ($14.4\pm5.8$ µg m$^{-2}$ yr$^{-1}$) (Larssen et al., 2008; Obrist et al., 2009; Fisher and Wolfe, 2012; Juillerat et al., 2012; Obrist et al., 2012; Risch et al., 2012; Benoit et al., 2013; Navrátil et al., 2014; Gerson et al., 2017; Risch et al., 2017; Risch and Kenski, 2018). Throughfall Hg deposition is another important way for Hg input in forests, Wright et al. (2016) summarized previous studies and reported the median throughfall Hg deposition to be 49.0, 16.3 and 7.0 µg m$^{-2}$ yr$^{-1}$ in Asia, Europe and North America, respectively. Large discrepancies in Asian co-located comparisons between rainfall and throughfall Hg depositions ($32.9\pm18.9$ and $13.3\pm8.6$ µg m$^{-2}$ yr$^{-1}$, respectively) could indicate a high dry deposition level in Asian forests (Wan et al., 2009; Wang et al., 2009; Fu et al., 2010a; Fu et al., 2010b; Luo et al., 2016; Ma et al., 2015; Han et al., 2016; Fu et al., 2016a; Ma et al., 2016; Wang et al., 2016b; Zhou et al., 2016)."

*Comment:*

Lines 276-278: Not sure how you generated this conclusion (see the Figure provided in Wright et al, 2016) on this topic.

*Response:*

We have revised this paragraph substantially. Please refer to Lines 243–251 in the revised manuscript:

"Throughfall Hg deposition is another important way for Hg input in forests, Wright et al. (2016) summarized previous studies and reported the median throughfall Hg deposition to be 49.0, 16.3 and 7.0 µg m$^{-2}$ yr$^{-1}$ in Asia, Europe and North America, respectively. Large discrepancies in Asian co-located comparisons between rainfall and throughfall Hg depositions ($32.9\pm18.9$ and $13.3\pm8.6$ µg m$^{-2}$ yr$^{-1}$, respectively) could indicate a high dry deposition level in Asian forests (Wan et al., 2009; Wang et al., 2009; Fu et al., 2010a; Fu et al., 2010b; Luo et al., 2016; Ma et al., 2015; Han et al., 2016; Fu et al., 2016a; Ma et al., 2016; Wang et al., 2016b; Zhou et al., 2016)."

*Comment:*

Line 313: Any reference to support this statement?

*Response:*

We have clarified the statement and added the citation. Please refer to Lines 270–275 in the revised manuscript:

"The measurements of precipitation volume by samplers have non-negligible uncertainties (Wetherbee, 2017). The relative standard deviations (RSDs) of daily and annual precipitation depth measurements in MDN were estimated to be 15 % and 10 %, respectively (Wetherbee et al., 2005). The event-based sampling volume biases of two types of samplers used in APMMN were estimated to be up to 11–18 % (Sheu et al., 2019)."

*Comment:*

Section 3.1.2: see comment above about evaporation of the non-precipitating events.

*Response:*

The uncertainty generated from the evaporation of the non-precipitation events is included in the uncertainty of cloud, fog, dew or frost deposition measurements.

*Comment:*

Section 3.2.1: There are some confusions here: Eq. (4) is the inferential method for estimating (modeling) dry deposition while the paragraph following this equation discusses flux measurement.

*Response:*

We have moved this equation to Section 4.2 to avoid the confusion. Please refer to Eq. (7) in Line 499 in the revised manuscript.

*Comment:*

Line 490: I cannot agree with the stamen, or this statement is not expressed clearly.

*Response:*

We have clarified this statement. Please refer to Lines 389–391 in the revised manuscript:

"In forest ecosystems, the presence of canopy changes the form of Hg deposition. The sum of litterfall and throughfall is more commonly used to represent the total Hg deposition in forests (Wang et al., 2016a; Wright et al., 2016)."

*Comment:*

Line 528: a portion of wet deposition may be intercepted by the canopy and cannot reach to the throughfall, especially if precipitation amount is small.

*Response:*

We have revised this statement. Please refer to Lines 426–428 in the revised manuscript:

"Throughfall Hg deposition includes the wet-deposited Hg passing through the canopy and a portion of dry-deposited Hg washed off from the canopy (Blackwell and Driscoll, 2015a; Wright et al., 2016)."

*Comment:*

Section 4: Are the whole section focusing on CTMS? If so, the title may specify the discussions are for CTMs

*Response:*

Section 4 is mainly talk about uncertainties in Hg deposition simulation using CTMs, but not the whole section. The model in Section 4.3 is not included in existing CTMs. Therefore, we tend to keep the current title.

*Comment:*

Paragraph starting on line 837: uncertainty discussion on GOM measurements should be combined into earlier sections. Here only list a brief recommendation for future research needs.

*Response:*

We have moved the uncertainty discussion to Section 4.2.1. Please refer to Lines 543–547 in the revised manuscript:

"It should be noted that the correction factor of 3 is not universally applicable. Different humidity levels or ozone concentrations lead to a significant change in underestimation. Different chemical forms of GOM also have different KCl capture efficiencies. Therefore, accurate quantification methods for measuring the total and chemically speciated GOM concentration are in urgent needs."

*Comment:*

Line 849: It is pretty clear that GEM dry deposition is more important than previously considered in CTMS over vegetated surfaces. It is a dominant contributor in the dry deposition budget (dry dep of GEM>GOM+PBM) over regions with low GEM emissions.

*Response:*

We have deleted this sentence.

*Comment:*

Line 857: Again, I feel such deposition may not necessarily stay on the ground surface.

*Response:*

This type of deposition has been reported in many existing studies. The uncertainty generated from the evaporation of the non-precipitation events is included in the uncertainty of cloud, fog, dew or frost deposition measurements.

*Comment:*

Line 866-867: not exactly true (in some parts of Asia).

*Response:*

We have revised this statement. Please refer to Lines 758–756 in the revised manuscript:

[revised manuscript text omitted]